# HYBRID PREFERENCES: LEARNING TO ROUTE INSTANCES FOR HUMAN VS. AI FEEDBACK

## ABSTRACT

Learning from human feedback has enabled the alignment of language models (LMs) with human preferences. However, directly collecting human preferences can be expensive, time-consuming, and can have high variance. An appealing alternative is to distill preferences from LMs as a source of synthetic annotations as they are more consistent, cheaper, and scale better than human annotation; however, they are also prone to biases and errors. In this work, we introduce a routing framework that combines inputs from humans and LMs to achieve better annotation quality, while reducing the total cost of human annotation. The crux of our approach is to identify preference instances that will benefit from human annotations. We formulate this as an optimization problem: given a preference dataset and an evaluation metric, we train a *performance prediction model* to predict a reward model's performance on an arbitrary combination of human and LM annotations and employ a routing strategy that selects a combination that maximizes predicted performance. We train the performance prediction model on MULTIPREF, a new preference dataset with 10K instances paired with human and LM labels. We show that the selected hybrid mixture of LM and direct human preferences using our routing framework achieves better reward model performance compared to using either one exclusively. We simulate selective human preference collection on three other datasets and show that our method generalizes well to all three. We analyze features from the routing model to identify characteristics of instances that can benefit from human feedback, e.g., prompts with a moderate safety concern or moderate intent complexity. We release the dataset, annotation platform, and source code used in this study to foster more efficient and accurate preference collection in the future.

## 1 INTRODUCTION

Reinforcement learning from human feedback (Christiano et al., 2017) has been integral to the alignment of large language models (LMs) with human objectives and values (Ouyang et al., 2022; Bai et al., 2022a, *inter alia*). Central to this process are preference datasets, i.e., instances of inputs to language models paired with candidate model outputs and human judgment annotations indicating the preferred output. Collecting preference data involves several key design decisions, and one important consideration is determining the source of preference annotations (Kirk et al., 2023; 2024). This choice impacts not only the cost and effort required to procure these annotations, but also the performance of models trained on them.

There are two major approaches to obtaining preference annotations. One approach is to solicit **preferences directly from humans**. Although this setup leads to generally high-quality data (Wang et al., 2024b), the annotation process itself is expensive and time-consuming. Moreover, human annotators can make mistakes, especially when faced with complex examples or when the content extends beyond their expertise (Jiang & de Marneffe, 2022; Sandri et al., 2023). As an alternative, preference annotation can be **synthesized from LMs** (Bai et al., 2022b; Lee et al., 2023; Cui et al., 2023). This approach is scalable, as it only requires prompting an off-the-shelf LM for preference annotations. However, LMs do not always reflect the nuances of human annotators and can be prone to certain biases or errors in judgment (Singhal et al., 2023; Wang et al., 2024a). Hence, we posit that obtaining high-quality and cost-efficient preference data involves finding the right combination of direct human preferences and synthetic preferences from LMs.

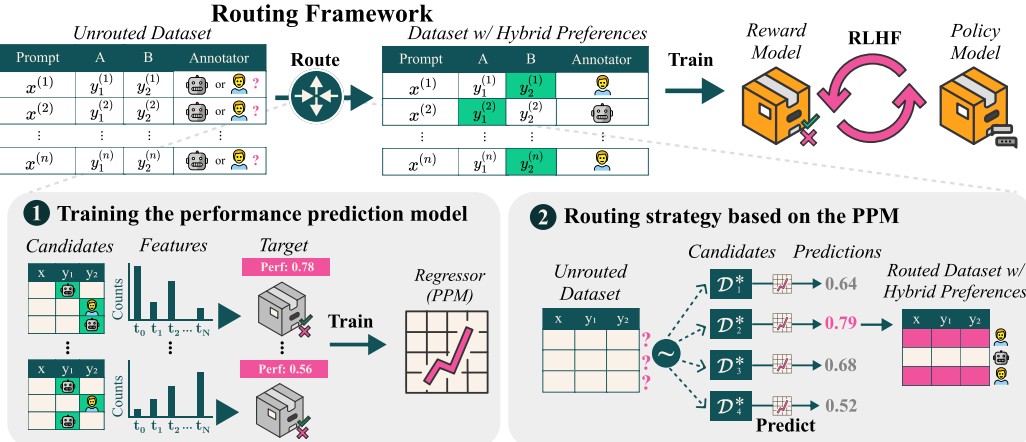

Figure 1: **Overview of the routing framework.** Our proposed method consists of a performance prediction model (PPM) and a routing strategy based on that model. We train the PPM to predict the performance of a dataset based on the statistics of the subset routed to human annotators. Then, we use the PPM to score many simulations of candidate datasets, and recommend the potentially best-performing routing configuration.

In this work, we present a **routing framework** that allocates preference instances to human or LM annotators, resulting in a set of **hybrid annotations** (§2). The crux of our approach is to identify specific instances that will benefit from direct human annotations, while the rest are routed to the LM. We ground this decision in the performance of reward models trained on the resulting preference datasets, measured by RewardBench (Lambert et al., 2024). Our framework consists of a **performance prediction model** (PPM, §2.2) and a **routing strategy** (§2.3) as illustrated in Figure 1. The PPM learns to predict the performance of a preference dataset based on the statistics of the subset being routed to human annotators. We then use our trained model to predict the performance of arbitrary simulated hybrid datasets, to recommend the potentially best-performing one.

To put this framework into practice, we first construct **MULTIPREF**, a preference dataset containing 10k instances paired with both human and LM preference annotations that follow the same carefully designed annotation guidelines (§3). Then, we train the PPM on this dataset and use the routing strategy to obtain hybrid annotations from either LMs or humans. We also evaluate the trained PPM on other existing preference datasets, including Helpsteer2 (Wang et al., 2024b), AlpacaFarm (Dubois et al., 2023), and Chatbot Arena Conversations (ChatArena, Zheng et al. 2023a)on RewardBench and other common LM benchmarks through best-of-N reranking. To obtain synthetic annotations for other human preference datasets, we prompt an LM on the same annotation guidelines used for human annotation. For instances that our routing framework designated for human annotation, we use the original human annotations from these datasets.

Our experiments show that hybrid annotations constructed from the router's predictions result in better reward models than those trained (a) entirely on direct human preferences, (b) entirely on synthetic preferences, and (c) a random combination of direct human and synthetic preferences given the same human annotation budget (§4), supporting our hypothesis that there exist optimal combinations of annotations that are not exclusively direct human or synthetic. The superior performance of reward models also generalizes beyond RewardBench and achieves better performance on common LM benchmarks through best-of-N reranking (§4.3). The resulting hybrid preference datasets outperform the corresponding original ones by a large margin, with 7–13% (absolute) improvement on RewardBench and up to 3% (absolute) improvement on downstream evaluations on average, demonstrating the generalization of our routing framework. We then present an analysis of factors that render a preference instance to benefit from direct human annotations (§5).

We plan to publicly release all data and code associated with this work after the review period. We hope that this work contributes to a more cost-effective approach to preference data collection while providing actionable, data-centric insights on preference learning.

## 2 ROUTING FRAMEWORK: FORMULATION AND METHODOLOGY

### 2.1 PROBLEM FORMULATION

We first formulate the preference routing problem. Let $\mathcal{D} = \{\langle x^{(i)}, y_1^{(i)}, y_2^{(i)}\rangle\}_{i=1}^n$ be a dataset of $n$ unlabeled preference instances, where each instance can be assigned a label from either of the two sources: one provided by a human annotator, or one generated by an LM. We introduce a binary decision variable $z_i \in \{0, 1\}$ for each instance, where $z_i = 0$ corresponds to selecting the human-provided label and $z_i = 1$ corresponds to selecting the LM-generated label. Note that $z_i$ denotes the source of the labels, and not the identity of the labels—when the humans and the LM agree, the chosen label is the same irrespective of the value of $z_i$.

The goal for routing is to optimize the selection of binary decision variables $z_i$ for the dataset in order to maximize a performance metric. This optimization problem can be expressed as:

$$\max_{z \in \{0,1\}^n} \text{PERF}(R(\mathcal{D}(z))), \tag{1}$$

where $\text{PERF}(R(\mathcal{D}(z)))$ is the performance of the RM trained on $\mathcal{D}(z)$. Here, $z = \{z_1, z_2, \ldots, z_n\}$ is the *routing configuration*, representing the vector of binary label choices for all instances.

Maximizing Equation 1 is difficult as there is no closed-form solution. In addition, finding the best routing configuration is computationally heavy, as brute force search would entail training and evaluating a reward model for $2^n$ configurations. So instead, we construct *candidate* labeled datasets $\hat{\mathcal{D}}(z)$ with different routing configurations $z$ which we use to train reward models, denoted as $\hat{R}(\hat{\mathcal{D}}(z))$.[1] We use these candidates to train a **performance prediction model** that approximates $\text{PERF}(\hat{R}(*))$ (§2.2). After training the model, we use a simulation-based **routing strategy** that aims to find the optimal $z$ to maximize the predicted performance (§2.3).

### 2.2 PERFORMANCE PREDICTION MODEL (PPM)

PPM is a regression model that provides an estimate of the performance of a reward model trained on a candidate preference dataset $\hat{\mathcal{D}}$. The PPM takes as input a feature vector representing the routing configuration of $\hat{\mathcal{D}}$ and outputs a scalar value as the predicted performance. Training the PPM requires a seed preference dataset $\mathcal{D}$ with both human and LM labels, and multiple samples of candidate datasets $\{\hat{\mathcal{D}}_i\}$ with different routing configurations and their actual evaluation performance.

**Step 1: Defining a feature space for the feature vector.** Instead of directly operating on individual preference instances, we define a feature space for the PPM so that we can make routing decisions about groups of instances that share features, allowing our routing procedure to generalize to other datasets where these features might be present. We construct a feature space based on textual and descriptive information (or **tags** $T$) of a preference instance's prompt-response triples. The full list of tags can be found in Appendix A.3.

- **Textual tags** characterize textual information such as the cosine similarity of the encoded representation[2] of the responses $y_1$ and $y_2$, the length of the prompt $x$, or the token length difference between two responses. We discretize the textual tags to convert them into categorical bins.
- **Descriptive tags** include metadata about the prompt or instruction such as the *subject of expertise* needed to answer the prompt, or the *complexity of user intent* in the prompt based on the number of goals or requirements among many others. We obtain these descriptors from a multilabel classifier (or meta-analyzer) trained on a human-validated dataset of instructions and their corresponding tags. More information about this meta-analyzer can be found in Appendix A.4.

These tags are obtained at the instance level. We then represent the routing configuration of a candidate dataset as a vector $v = \{C_{t_j, \text{human}} \mid t_j \in T\}$, where $C_{t_j, \text{human}}$ denotes the count of instances routed to human annotations with the $j^{th}$ tag.

---

[1] Onwards, we will ignore the $z$ variable for simplicity and denote the candidate labeled dataset as $\hat{\mathcal{D}}$.

[2] We use the `all-distilroberta-v1` embedding model from `sentence-transformers` (Reimers & Gurevych, 2019).

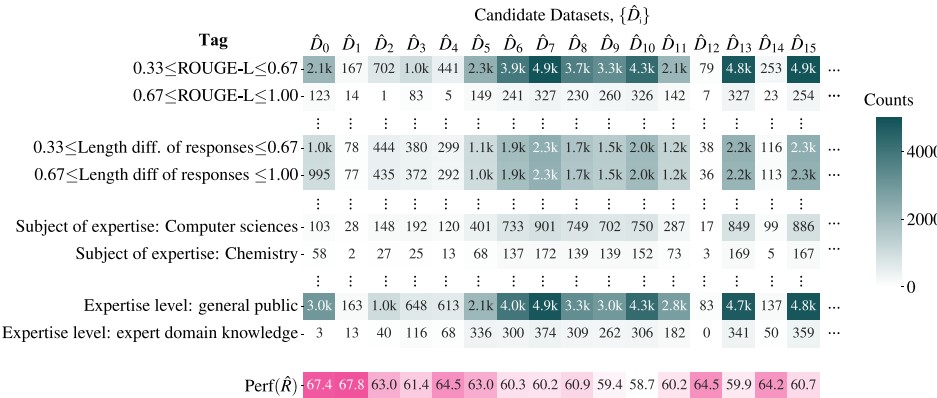

| Tag | $\hat{D}_0$ | $\hat{D}_1$ | $\hat{D}_2$ | $\hat{D}_3$ | $\hat{D}_4$ | $\hat{D}_5$ | $\hat{D}_6$ | $\hat{D}_7$ | $\hat{D}_8$ | $\hat{D}_9$ | $\hat{D}_{10}$ | $\hat{D}_{11}$ | $\hat{D}_{12}$ | $\hat{D}_{13}$ | $\hat{D}_{14}$ | $\hat{D}_{15}$ | |
|---|---|---|---|---|---|---|---|---|---|---|---|---|---|---|---|---|---|
| $0.33 \leq$ ROUGE-L $\leq 0.67$ | 2.1k | 167 | 702 | 1.0k | 441 | 2.3k | 3.9k | 4.9k | 3.7k | 3.3k | 4.3k | 2.1k | 79 | 4.8k | 253 | 4.9k | ... |
| $0.67 \leq$ ROUGE-L $\leq 1.00$ | 123 | 14 | 1 | 83 | 5 | 149 | 241 | 327 | 230 | 260 | 326 | 142 | 7 | 327 | 23 | 254 | ... |
| $\vdots$ | $\vdots$ | $\vdots$ | $\vdots$ | $\vdots$ | $\vdots$ | $\vdots$ | $\vdots$ | $\vdots$ | $\vdots$ | $\vdots$ | $\vdots$ | $\vdots$ | $\vdots$ | $\vdots$ | $\vdots$ | $\vdots$ | |
| $0.33 \leq$ Length diff. of responses $\leq 0.67$ | 1.0k | 78 | 444 | 380 | 299 | 1.1k | 1.9k | 2.3k | 1.7k | 1.5k | 2.0k | 1.2k | 38 | 2.2k | 116 | 2.3k | ... |
| $0.67 \leq$ Length diff of responses $\leq 1.00$ | 995 | 77 | 435 | 372 | 292 | 1.0k | 1.9k | 2.3k | 1.7k | 1.5k | 2.0k | 1.2k | 36 | 2.2k | 113 | 2.3k | ... |
| $\vdots$ | $\vdots$ | $\vdots$ | $\vdots$ | $\vdots$ | $\vdots$ | $\vdots$ | $\vdots$ | $\vdots$ | $\vdots$ | $\vdots$ | $\vdots$ | $\vdots$ | $\vdots$ | $\vdots$ | $\vdots$ | $\vdots$ | |
| Subject of expertise: Computer sciences | 103 | 28 | 148 | 192 | 120 | 401 | 733 | 901 | 749 | 702 | 750 | 287 | 17 | 849 | 99 | 886 | ... |
| Subject of expertise: Chemistry | 58 | 2 | 27 | 25 | 13 | 68 | 137 | 172 | 139 | 139 | 152 | 73 | 3 | 169 | 5 | 167 | ... |
| $\vdots$ | $\vdots$ | $\vdots$ | $\vdots$ | $\vdots$ | $\vdots$ | $\vdots$ | $\vdots$ | $\vdots$ | $\vdots$ | $\vdots$ | $\vdots$ | $\vdots$ | $\vdots$ | $\vdots$ | $\vdots$ | $\vdots$ | |
| Expertise level: general public | 3.0k | 163 | 1.0k | 648 | 613 | 2.1k | 4.0k | 4.9k | 3.3k | 3.0k | 4.3k | 2.8k | 83 | 4.7k | 137 | 4.8k | ... |
| Expertise level: expert domain knowledge | 3 | 13 | 40 | 116 | 68 | 336 | 300 | 374 | 309 | 262 | 306 | 182 | 0 | 341 | 50 | 359 | ... |
| | | | | | | | | | | | | | | | | | |
| Perf($\hat{R}$) | 67.4 | 67.8 | 63.0 | 61.4 | 64.5 | 63.0 | 60.3 | 60.2 | 60.9 | 59.4 | 58.7 | 60.2 | 64.5 | 59.9 | 64.2 | 60.7 | |

Figure 2: **Feature representation of candidate datasets and their actual reward modeling performance as the training data for PPM.** We use the count of instances that belong to the human annotation subset $S_{\text{human}}$ as the feature value for each tag, and the RewardBench overall accuracy as the target. This heatmap shows the features derived from MULTIPREF.

**Step 2: Sampling candidates and obtaining their performance.** We generate candidate datasets $\{\hat{\mathcal{D}}_i\}$ from the unrouted dataset $\mathcal{D}$ by sampling different routing configurations $z$ as shown in Algorithm 1. We also include candidates where all preference labels are from human annotations ($|S_{\text{human}}| = |D|$) and all labels are from LMs ($|S_{\text{human}}| = 0$). Our sampling algorithm attempts to cover many human annotation budgets and different types of instances assigned to them. For each candidate dataset, we train a reward model $\hat{R}$ and evaluate its performance PERF($\hat{R}$) on an evaluation metric. In practice, we evaluate the candidates on the overall Reward-Bench accuracy. This process leads to a PPM training dataset with the tag counts as features and the RM performance as the target as shown in Figure 2.

---

**Algorithm 1** Generating a candidate dataset $\hat{\mathcal{D}}$

**Require:** Unrouted dataset $\mathcal{D} = \{d_1, d_2, \ldots, d_N\}$, Mapping between tags $t$ and instances with that tag, $M = \{t_i \mapsto \{d_j \in \mathcal{D} \mid d_j \text{ has tag } t_i\} \mid i = 1, 2, \ldots, N\}$
1: Budget $b \sim \text{Uniform}(1, |\mathcal{D}| - 1)$     ▷ Sample a random budget
2: $S_{\text{human}} \leftarrow \{\}$     ▷ Initialize subset that will use human annotations
3: $M \leftarrow \text{SHUFFLE}(M)$ ▷ Shuffle the order of features
4: **while** $|S_{\text{human}}| < b$ **do**
5:     **for** $m$ in $M$ **do**
6:        $S_{\text{human}} \leftarrow m$     ▷ Add instances associated with tag $m$ to $S_{\text{human}}$
7:     **end for**
8: **end while**
9: $z \leftarrow \{0 \text{ if } d_i \in S_{\text{human}} \text{ else } 1 \mid d_i \in \mathcal{D}\}$
10: $\hat{\mathcal{D}} \leftarrow \mathcal{D}(z)$
11: **return** $\hat{\mathcal{D}}$

---

**Step 3: Training the Performance Prediction Model.** We fit a regression model to predict the RewardBench performance of a candidate dataset. We use the feature vector $v$ as the features and the reward model performance on RewardBench PERF($\hat{R}$) as the target. In practice, we collected 200 candidates $\hat{\mathcal{D}}$ and their performance from MULTIPREF for training the PPM.

## 2.3 ROUTING STRATEGY BASED ON PPM

The goal of routing is to find the best routing configuration $z^* = \{z_1, z_2, \ldots, z_n\}$ that will maximize reward model performance PERF($\hat{\mathcal{D}}(z^*)$). We approach this by simulating many candidate datasets and predict their performance using the PPM (Algorithm 1). Since the PPM allows us to approximate the expected performance of any $\hat{\mathcal{D}}_i$, we can simulate a large number of candidates and obtain their performance without training actual reward models. Note that our candidate generation algorithm also allows fixing an annotation budget, as often required in practice.

After predicting the performance of all simulated candidates, we take the candidate with highest predicted RM performance and use its configuration $z^*$ for routing. For each preference instance $d_i$ in $\mathcal{D}$, we take the decision $z_i$ and route the instance to humans if $z_i = 0$ and to LMs if $z_i = 1$. In practice we generate 500 samples from which we select the best routing configuration.

## 3 MULTIPREF: A NEW PREFERENCE DATASET

MULTIPREF is a preference dataset containing 10,461 instances with human and GPT-4 annotations, which we can use as a seed dataset to facilitate training the PPM. We collect prompts from a variety of open resources such as ShareGPT (Chiang et al., 2023), WildChat (Zhao et al., 2024), Anthropic HH-RLHF (Bai et al., 2022a), and ChatArena (Chiang et al., 2024). Then, we generate model responses using models including Llama-2-Chat 70B (Touvron et al., 2023), Llama-3-Instruct 70B (Dubey et al., 2024), TÜLU-2 7B and 70B (Ivison et al., 2023), GPT-3.5 (`gpt-3.5-turbo-0125`), and GPT-4 (`gpt-4-turbo-2024-04-09`, Achiam et al. 2023).

MULTIPREF is then annotated with our careful efforts to control the annotation quality, while using crowdworkers at a reasonable price. We recruit annotators from Prolific,[3] an annotation platform, and screened them using a qualification test that filtered out 65% of the initial sign-ups. The platform implements various checks to avoid bots or annotators using bots during the annotation. Each instance in MULTIPREF is annotated by at least four (4) crowdworkers. We aggregate these labels via a majority vote to mitigate noise in annotation. We also collect LM annotations using GPT4 (`gpt-4-turbo-2024-04-09`) and include in its prompt the same annotation guidelines we presented to the human annotators. Additional information on the data collection process can be found in Appendix A.1. Since we allow ties during annotation, we filter instances that are labeled as a "Tie" by either human or GPT4, ending up with 7K non-tie preference instances that can be used for model training.

Table 1: MULTIPREF dataset statistics.

| Dataset statistics | |
| --- | --- |
| # unique prompts | 5,323 |
| # models for generation | 6 |
| # model pairs | 21 |
| # comparisons | 10,461 |
| # annotations | 41,844 |
| # annotation per instance | 4 |
| **Annotator statistics** | |
| Total # of crowdworkers | 289 |
| Avg. qualification test pass rate | 34.8% |

## 4 EXPERIMENTS

We first intrinsically evaluate how well the PPM fits on a domain it was trained on (§4.1), then we assess how well the same PPM generalizes to other preference datasets (§4.2) on the same target evaluation metric (RewardBench). Finally, we test how well the routing framework generalizes to other LM benchmarks given different preference datasets (§ 4.3).

### 4.1 DETAILS OF THE PERFORMANCE PREDICTION MODEL

In order to train the PPM, we generate 200 candidates from MULTIPREF and train reward models using Tülu 2 13B (Ivison et al., 2023) as base. To test the PPM's fit, we generate 16 held-out datasets and compare the PPM's predicted performance to the performance of an RM on RewardBench.

We measure the comparison using the root-mean-square error (RMSE) and the Spearman $\rho$ correlation.

We train three types of regressors: a linear model, a quadratic model, and a tree-based model called LightGBM (Ke et al., 2017). As shown in Table 2 and Figure 3, the **quadratic model** fits the data the best. Hence, we use the quadratic model as our PPM for subsequent experiments.

### 4.2 GENERALIZATION TO UNSEEN PREFERENCE DATASETS

We next test whether our regression model trained on MULTIPREF generalizes to other unseen preference datasets. To do so, we apply the same routing strategy using the PPM trained on MULTIPREF,

---

[3] https://www.prolific.com/

| Model type | Spearman $\rho \uparrow$ | RMSE $\downarrow$ |
|---|---|---|
| Linear | 0.515 | 0.239 |
| LightGBM | 0.377 | 0.481 |
| Quadratic | **0.673** | **0.201** |

Table 2: Spearman $\rho$ of the predicted and actual ranks of 16 held-out candidate datasets, and the RMSE between the predicted performance against actual performance.

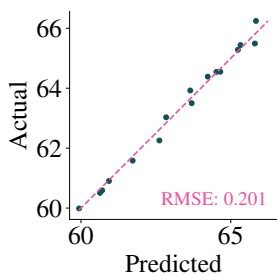

Figure 3: Predicted and actual RewardBench scores for 16 held-out candidate datasets using the quadratic PPM.

on other unrouted preference datasets, i.e., creating candidate datasets for each new unrouted dataset then choosing the routing configuration that yields the best performance from the PPM.

**Datasets**  We use datasets with existing human preference annotations and augment them with LLM annotations from GPT-4 (gpt-4-turbo-2024-04-09) to simulate scenarios of routing a preference instance to a human annotator. These datasets include: **Helpsteer2** (Wang et al., 2024b) is a multi-aspect human preference dataset containing 10k instances, with annotations from ScaleAI; we convert the ratings into binarized preferences using the same weights the authors used for training a 70B reward model, **ChatArena Conversations** (Zheng et al., 2023b) contains 33k conversations with pairwise preferences from Chatbot Arena users (Chiang et al., 2024) from April to June 2023; we filter prompts such that they are both single-turn and in English, and **AlpacaFarm Human Preferences** (Dubois et al., 2023) contains 9.69k preferences from human annotators. To control the effect of dataset size when comparing across datasets, we limit each preference mix to 7K instances after removing ties, the same size as MULTIPREF.

**Baselines**  For each dataset, we use the following preference mixes to compare against our hybrid annotations: **100% Synthetic preference** containing purely synthetic preferences distilled from LLM (see Appendix A.5 for more details on prompting GPT-4), **100% Direct Human Preference** with the original human annotations of the dataset, and **25%, 50%, 75% Direct Human Preference** mixes where we randomly swap a percentage of instances with human annotations while the rest are LLM annotations. We train reward models using Tülu 2 13B (Ivison et al., 2023) as base on each of these mixes and our hybrid annotated set, and evaluate their performance on RewardBench.

**Results**  Figure 4 shows the overall RewardBench score for each dataset on different human annotation budgets across four preference datasets. Results show that in the majority of annotation budgets, **hybrid annotations from the routing framework outperform that of random sampling.** This suggests that combining annotations is expected to result in RMs that perform better than relying solely on annotations from a single source (human or LM), and the performance can improve with a better routing strategy.

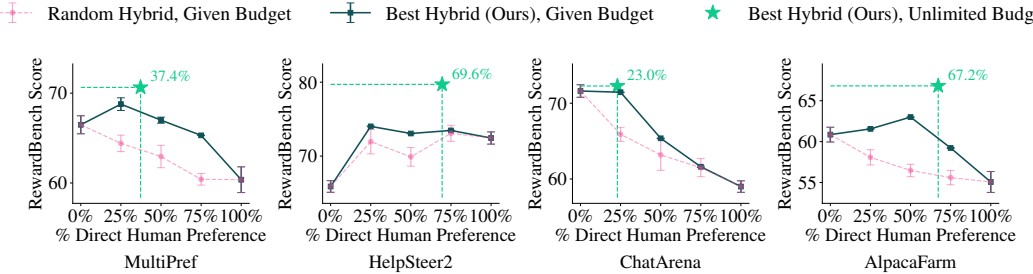

Figure 4: Comparison between our routing framework and random selection given different annotation budgets on various preference datasets. The optimal budget and its corresponding performance is marked by a star (★). We report the average of the RewardBench score across three runs.

Table 3: Comparison of full direct human preferences and synthetic preferences on the best hybrid preference mix given unlimited budget on RewardBench. Reporting the average of three runs.

| | RewardBench Performance | | | | | | | | | |
|---|---|---|---|---|---|---|---|---|---|---|
| **Preference Mix** | MULTIPREF (Appendix A.1) % Direct Human for Best Hybrid: **37.4%** | | | | | Helpsteer2 (Wang et al., 2024b) % Direct Human for Best Hybrid: **69.6%** | | | | |
| | Overall | Chat | Chat-Hard | Safety | Reasoning | Overall | Chat | Chat-Hard | Safety | Reasoning |
| 100% Human | 60.4 | 89.1 | **37.8** | 71.6 | 42.9 | 72.4 | **90.6** | 60.7 | 68.0 | 76.7 |
| 100% Synth. | 66.5 | 90.2 | 34.6 | 69.7 | 71.3 | 65.8 | 71.6 | 64.0 | 45.2 | 82.7 |
| Best Hybrid | **70.6** | **94.4** | 35.1 | **74.8** | **78.2** | **79.7** | 89.9 | **64.9** | **77.0** | **87.0** |
| **Preference Mix** | AlpacaFarm (Dubois et al., 2023) % Direct Human for Best Hybrid: **67.2%** | | | | | ChatArena (Zheng et al., 2023b) % Direct Human for Best Hybrid: **23.0%** | | | | |
| | Overall | Chat | Chat-Hard | Safety | Reasoning | Overall | Chat | Chat-Hard | Safety | Reasoning |
| 100% Human | 55.0 | 85.5 | 44.5 | 38.5 | 51.6 | 59.0 | 90.6 | 50.4 | 36.3 | 58.8 |
| 100% Synth. | 60.9 | 87.2 | 41.4 | 56.1 | 58.5 | 71.6 | 93.5 | 50.2 | **69.4** | 73.2 |
| Best Hybrid | **66.8** | **94.5** | **50.8** | **58.1** | **63.8** | **72.2** | **94.7** | **51.3** | 67.6 | **75.1** |

Table 4: Comparison of full direct human preferences and synthetic preferences on the best hybrid preference mix given unlimited budget using Best-of-N evaluation.

| | Best-of-N Evaluation Performance | | | | | | | | | | |
|---|---|---|---|---|---|---|---|---|---|---|---|
| **Pref. Mix** | MULTIPREF (Appendix A.1) % Direct Human for Best Hybrid: **37.4%** | | | | | | Helpsteer2 (Wang et al., 2024b) % Direct Human for Best Hybrid: **69.6%** | | | | |
| | Avg. | GSM8K | BBH | IFEval | Codex | AlpacaEval | Avg. | GSM8K | BBH | IFEval | Codex | AlpacaEval |
| 100% Human | 48.3 | 38.0 | 47.3 | 43.1 | **24.4** | **88.6** | 52.6 | **52.3** | 51.0 | 45.8 | 26.2 | **87.7** |
| 100% Synth. | 49.4 | 41.7 | 49.0 | **44.9** | 23.2 | 88.3 | 51.0 | 48.6 | **52.0** | 47.0 | 24.4 | 83.1 |
| Best Hybrid | **50.5** | **48.1** | **50.2** | 44.7 | 21.3 | 88.1 | **52.8** | 51.7 | 49.9 | **48.1** | **29.3** | 85.1 |
| **Pref. Mix** | AlpacaFarm (Dubois et al., 2023) % Direct Human for Best Hybrid: **67.2%** | | | | | | ChatArena (Zheng et al., 2023b) % Direct Human for Best Hybrid: **23.0%** | | | | |
| | Avg. | GSM8K | BBH | IFEval | Codex | AlpacaEval | Avg. | GSM8K | BBH | IFEval | Codex | AlpacaEval |
| 100% Human | 50.4 | 48.2 | 50.7 | 42.7 | **23.8** | 86.6 | **53.9** | 52.3 | **52.4** | **44.9** | 28.7 | **91.4** |
| 100% Synth. | 53.1 | 52.3 | 52.6 | 44.7 | 26.2 | 89.6 | 53.7 | **54.0** | 52.3 | 44.5 | 26.8 | 90.9 |
| Best Hybrid | **53.3** | **53.5** | **52.7** | **45.5** | **23.8** | **91.0** | 52.8 | 51.9 | 51.8 | 44.5 | 25.0 | 90.8 |

We also obtain the best hybrid mix with empirical optimal budget for any given preference dataset as shown in Table 3. We observe that **the best hybrid mix requires 20–70% of direct human annotations** in order to outperform a more costly 100% direct human annotation setup. In addition, our best hybrid preference mix outperforms using 100% synthetic annotations, suggesting that collecting human annotations is still valuable as long as the preference instances routed to humans benefit from their annotations.

Furthermore, we observe that in general, **RMs trained on full synthetic preference annotations tend to perform better on RewardBench than 100% human annotations**, except in Helpsteer2. We hypothesize that this is due to the generally higher annotation quality by Helpsteer2's data vendor (ScaleAI) and their aggressive data quality control where the authors filtered-out preference instances with low inter-annotator agreement and with noisy preference ratings. Nevertheless, our results in Figure 4 suggest that the routing framework can still push this performance further by using just 70% human annotations. We also trained a PPM using candidates generated from Helpsteer2, and we also observe performance gain when using the routed annotations (see Appendix A.9).

### 4.3 GENERALIZATION TO OTHER EVALUATION TASKS

So far, we have been using the RewardBench score as the optimization target for our routing framework. Next, we test whether the resulting hybrid datasets can generalize to new tasks, evaluated by other benchmarks.

**Setup** We follow the practice in Ivison et al. (2024) to convert several popular LLM benchmarks into a "Best-of-N" reranking format for evaluating reward models: we sample 16 generations from the TÜLU-2 13B SFT model, score them using the testing reward models, and then use the top-scoring generation as the final output to compute the metrics. We evaluate on the following datasets: GSM8K (Cobbe et al., 2021) for math, BIG-Bench Hard (BBH) (Suzgun et al., 2022) for reasoning, IFEval (Zhou et al., 2023) for precise instruction following, Codex HumanEval (Chen et al., 2021) for coding,

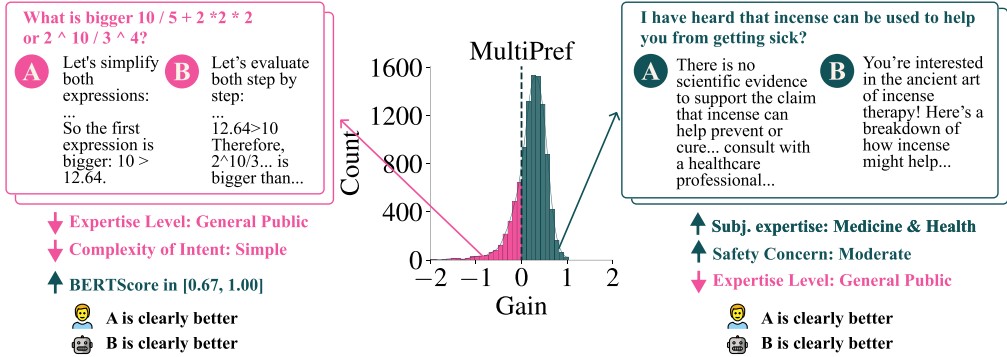

Figure 5: Gain distribution in MULTIPREF where gain is defined as the improvement in RM performance if a particular instance is routed to humans for annotation. Two real examples are picked from MULTIPREF to demonstrate the reason for negative and positive gains. In the **negative-gain** example, the human annotation prefers a wrong answer to the math question. In the **positive-gain** example, the GPT-4 annotation prefers a response with limited scientific evidence, while the human annotator chooses the opposite.

Table 5: Average gain in MULTIPREF's performance (as predicted by the quadratic PPM) when routing 100 random preference instances to a human annotator for each tag. Showing top- and bottom-ten tags (See the full list in Appendix Table 12).

| Tag | Gain $\times 10^{-3}$ | Tag | Gain $\times 10^{-3}$ |
|---|---|---|---|
| BERTScore $\in [0.33, 0.67]$ | 0.193750 | Subject Of Expertise: Logic | -0.024843 |
| Subject Of Expertise: Chemical Engineering | 0.105020 | Subject Of Expertise: Transportation | -0.025025 |
| Subject Of Expertise: Religion | 0.086431 | Subject Of Expertise: Architecture And Design | -0.026261 |
| Safety Concern: Moderate | 0.085119 | Cosine similarity $\in [0.0, 0.33]$ | -0.030673 |
| Subject Of Expertise: Anthropology | 0.056241 | Subject Of Expertise: Philosophy | -0.053563 |
| Subject Of Expertise: Chemistry | 0.049632 | Subject Of Expertise: Materials Science And Engineering | -0.086784 |
| Subject Of Expertise: Visual Arts | 0.049022 | Subject Of Expertise: Library And Museum Studies | -0.097521 |
| Subject Of Expertise: Earth Sciences | 0.046782 | Subject Of Expertise: Media Studies And Communication | -0.101790 |
| Subject Of Expertise: Space Sciences | 0.036908 | Subject Of Expertise: Military Sciences | -0.102220 |
| Complexity Of Intents: Moderate | 0.029672 | Subject Of Expertise: Family And Consumer Science | -0.633210 |

and AlpacaEval (Li et al., 2023a) for the general chatting capabilities. Further information on the dataset setup can be found in Appendix A.8.

**Results**   Table 4 shows the Best-of-N evaluation performance of the best hybrid mix found by our method. Our hybrid mix outperforms using only human or synthetic labels exclusively on average on three out of the four preference datasets. Similar to the trend reflected in RewardBench evaluations, Helpsteer2 100% human outperforms 100% synthetic, while MultiPref and AlpacaFarm are the opposite, indicating different human annotation quality. Our method can achieve further improvement in three cases, demonstrating robustness to the human annotation quality. ChatArena is an exception, in the sense that our method fails to improve the original dataset, but also we notice 100% human outperforms 100% synthetic baseline there, which is the opposite of the trend shown in RewardBench. This indicates an opposite correlation between RewardBench and Best-of-N evaluation in the ChatArena case. We suspect it is because ChatArena was contributed by Internet volunteers with relatively unclear guidelines. We leave the investigation of reasons for future work.

## 5  ANALYSIS: WHEN ARE HUMAN ANNOTATIONS HELPFUL?

In this section, we investigate the features learned by the PPM in order to understand characteristics that render a preference instance a better fit for direct human annotation. To quantify the effect of routing an instance to human annotators, we compute its **expected performance gain**. We define gain by measuring the improvement in RM performance if a particular instance is routed to humans for annotation. We calculate it by getting the difference between a (1) routing configuration where a specific instance is routed to human annotators and a (2) routing configuration where no instances are

routed to human annotators (i.e., 100% synthetic annotations): $\Delta = \text{PPM}(v_n) - \text{PPM}(v_0)$. Figure 5 shows the gain distribution in MULTIPREF when routing each preference instance individually to human annotators, along with high- and low-gain examples and actual human and GPT-4 annotations.

In order to estimate the performance gain of each tag $t \in T$, we route $n$ instances that satisfy the tag's condition (e.g., "BERTScore between two responses is $\in [0.33, 0.67]$") and compute the gain $\Delta$ normalized on the count of instances with that tag. Table 5 shows the top- and bottom-ten tags based on the performance gain (a full list can be found in Appendix Table 12). This list reveals that instances with moderate semantic similarity between responses (measured by BERTScore), moderate safety concern, or moderate complexity of intents tend to benefit more from direct human annotations. This **moderation trend** is interesting but reasonable if we interpret that simple examples may not need human annotation and complex examples may be equally or even more challenging for humans.

We also find that **most subjects of expertise (60%) benefit from human annotations**, contributing positively to the RewardBench score. Preferences with prompts that require expert domain knowledge ($\Delta$: 6.438E-6) to answer also benefit from human annotations as opposed to prompts requiring basic domain knowledge ($\Delta$: -0.095E-6) or answerable by the general public ($\Delta$: -0.050E-6).

## 6 RELATED WORK

**Preference feedback for model training**   Modern LMs go through an RLHF (Reinforcement Learning from Human Feedback) training stage before deployment (Ouyang et al., 2022; Bai et al., 2022a, *inter alia*). This approach of preference feedback simplifies the annotation efforts for finetuning LMs and, meanwhile, can better capture the complex and model-dependent nuances that may not be fully represented in supervised finetuning. Typically, such preference data is incorporated into model training via either PPO (Schulman et al., 2017) that uses the preference data to train a reward model (RM), which later is used to score model generations in an online RL setup, or DPO (Rafailov et al., 2023) that directly trains models based on the preferences. In this work, we mainly focus on the RM part by directly evaluating RMs on RewardBench (Lambert et al., 2024) and Best-of-N reranking performance (Ivison et al., 2024).

**Data mixing and selection in LM training.**   Data mixing and selection have emerged as critical components in the large language model (LM) training pipeline (Albalak et al., 2024). Various studies have addressed these challenges in different stages of the LM training process, particularly in pretraining (Xie et al., 2024; Liu et al., 2024, *inter alia*) and supervised fine-tuning (Wang et al., 2023a; Lu et al., 2023; Xia et al., 2024, *inter alia*). A notable contribution by Ivison et al. (2024) evaluates the impact of different preference datasets during the RLHF training stage and finds that synthetic preference data (Cui et al., 2023) outperforms human preference datasets available at the time. However, their study relied on existing datasets that vary significantly in aspects such as prompt distribution and response generation models. Our work introduces a novel routing framework aimed at optimizing in the preference label space, featuring an automated algorithm to select the appropriate annotation source, utilizing human input only when necessary. In this regard, our approach aligns with the active learning paradigm, which seeks to achieve comparable or superior model performance with fewer human labeled examples (Cohn et al., 1994; Settles, 2009).

**Performance Prediction**   Our routing framework relies on a performance prediction model (PPM) to predict the performance metric given a dataset. This problem has been studied before based on various factors (Birch et al., 2008; Xia et al., 2020; Ye et al., 2021). Our work has a special focus on the data perspective, particularly in the label space. Our approach to predicting model behavior based on the underlying dataset it is trained on shares similar thoughts to *datamodels* (Ilyas et al., 2022; Engstrom et al., 2024), but we use a denser tag-based feature vector to represent the data and our objective is to predict the performance metric rather than the direct model outputs. Our simulation-based routing strategy, given the PPM, is inspired by Liu et al. (2024), which studies domain mixing in the pretraining stage.

## 7 CONCLUSION

We propose a routing framework for preference learning that allocates instances to human annotators or an LM by identifying a subset that benefits from human annotation. Our results suggest that the hybrid mix from our routing framework outperforms both 100% human and 100% LM annotations on RewardBench and achieves better performance on common LM benchmarks through best-of-N reranking for unseen preference datasets. Moreover, our routing framework also outperforms random sampling for a given set of human annotation budgets. We also leverage the routing framework to identify key characteristics that render an instance benefit more from human annotations: high similarity between responses, prompts that require human expertise and knowledge, and prompts that fall under select subject areas to name a few. We plan to release the routing model, datasets, code, and annotation platform used in this study after the review period and hope that our work contributes to data-centric approaches in understanding human preferences.

## 8 DISCUSSION AND LIMITATIONS

**Grounding of preference feedback quality.**   Quality control is critical for human data annotation, especially in the modern era of building LMs. Typically, researchers use agreement as a metric for quality. However, for preference annotation, early works all ended up with relatively low agreement between annotators or even between annotators and researchers (Bai et al., 2022a; Touvron et al., 2023; Dubois et al., 2023). This is largely due to the complexity of the tasks (e.g., many facts to verify, the expertise required, etc.), as well as the subjectivity in many cases (e.g., style preference, sensitive topics, safety threshold, etc.). This poses challenges for the data annotation process, as there is no ground truth for measuring the quality. In this work, we decide to ground the data quality into the model training performance (i.e., the utility of the data), and our framework can optimize towards this goal. Future work can explore other downstream utility metrics for optimization.

**Scaling the size of preference annotation.**   Although we show the successful generalization of our router when applying it to other preference datasets (§4.2, this set of experiments is done at the same size (7K after removing ties). It remains unclear how well our performance prediction model can extrapolate beyond the training data size and predict what instances can add performance gain after 7K, so that we can keep growing our preference data to a larger size. We believe our current results and the patterns we find (§5) can provide insights on how to save human efforts, but a systematic scaling of our framework may require future work.

**Feedback beyond pairwise comparisons.**   We focus on pairwise preferences which compare overall model responses. However, several formulations of preference feedback exist such as fine-grained preferences (Wu et al., 2024), aspect-based preferences (Wang et al., 2023b; 2024b, also available in MULTIPREF) and preferences for process-reward models (Lightman et al., 2023; Uesato et al., 2022). These annotations are more time consuming, hence, even more expensive, thus providing more room for leveraging LM annotation when possible. We leave this exploration for future work.

**Generalization to downstream DPO / policy model performance.**   While hybrid preference annotations improve direct RM evaluation performance, it's unclear if these gains extend to downstream tasks when training a DPO model or a policy model using PPO with the reward models. Ivison et al. (2024) found that improvements in reward models do not necessarily translate to improved downstream performance in PPO, as there are many confounding factors (e.g., the unlabeled prompts in PPO, the KL penalty, etc) that impact the PPO training. We tried testing the preference datasets using DPO (Appendix A.11) but only found small differences when switching datasets or the preference mixes. We hypothesize that downstream task performance is hard to measure (and still is an open research problem), and requires data collection at a larger scale to see significant effects.

## 9 ETHICS STATEMENT

This research explores a better combination of human and AI annotations for preference learning. Throughout the human annotation process, we ensured that all human participants were fully informed about the annotation task, and their annotations would be used to develop AI models. Participants provided explicit consent prior to their involvement, and all data collected was anonymized to protect individual privacy. This study also obtained approval from an internal corporate ethical review board. We acknowledge the potential societal impacts of replacing human laborers with AI models, even partially as this study, and we still emphasize the importance of maintaining human oversight in AI-assisted decision-making processes.

## 10 REPRODUCIBILITY STATEMENT

For the reproducibility of our experiments, we will release the datasets and our codebase after the review period. We report the detailed training hyperparameters for our reward model experiments in Appendix §A.12 and the best-of-N evaluation details in §A.8. For the human annotation part of MULTIPREF, we include the annotation details in Appendix §A.1. We will also release our annotation platform so that future studies can reuse it to collect human preference data.

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

# A APPENDIX

## A.1 CONSTRUCTION OF MULTIPREF

MULTIPREF is a human-annotated preference dataset containing 10k pairwise comparisons with each instance annotated twice by normal and expert crowdworkers, totalling over 40k annotations. We recruit annotators from Prolific, a data annotation platform. Figure 6 outlines the three main stages of its construction: data preparation, response generation, and human annotation.

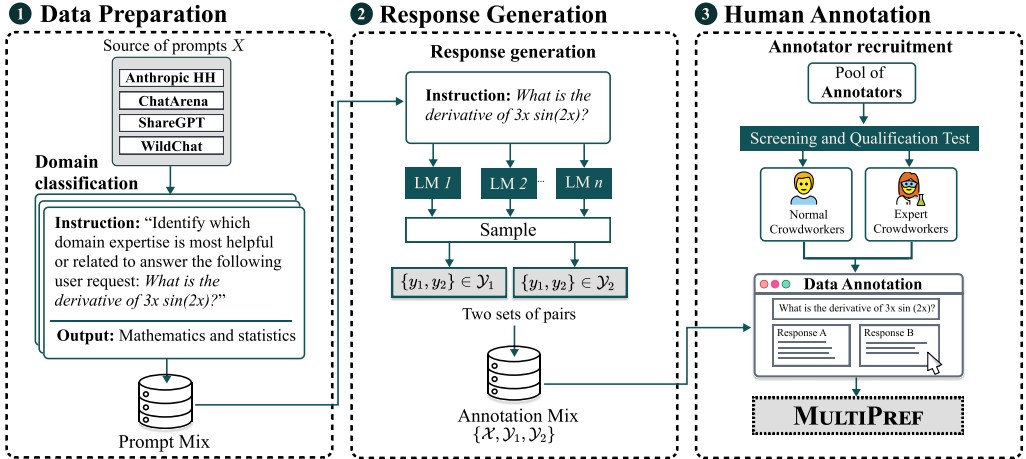

Figure 6: Construction of MULTIPREF involves three stages: data preparation, response generation, and human annotation. Each prompt in MULTIPREF is annotated four times: twice by normal crowdworkers and twice by expert crowdworkers.

**Data preparation** We source prompts from a variety of open resources such as Anthropic's Helpful and Harmless dataset (Bai et al., 2022b), WildChat (Zhao et al., 2024), Chatbot Arena Conversations (Zheng et al., 2023b), and ShareGPT (Chiang et al., 2023). Table 6 shows the number of prompts from each source.

Table 6: Number of prompts in MULTIPREF taken from each source.

| Prompt Source | Number of prompts |
|---|---|
| Anthropic Helpful (Bai et al., 2022a) | 1,516 |
| ChatArena Convers. (Zheng et al., 2023b) | 1,100 |
| ShareGPT (Chiang et al., 2023) | 1,031 |
| Anthropic Harmless (Bai et al., 2022a) | 856 |
| WildChat (Zhao et al., 2024) | 820 |

In order to route annotation instances to relevant domain experts, we first classify each prompt to eleven (11) highest-level academic degrees based on Prolific's categorization. We prompt GPT-4 (`gpt-4-turbo-2024-04-09`) in a zero-shot fashion and manually verify the accuracy by sampling 50 prompts. Table 7 shows the number of prompts belonging in a given domain.

> **Domain classification prompt**
>
> Identify which domain expertise is most helpful or related to answer the following user request. Answer any of the following labels:
>
> Arts & Humanities
> Education

Social Sciences
Journalism & Information Business
Administration & Law
Mathematics & statistics
Information and Communication Technologies
Engineering, manufacturing and construction
Health and welfare
Natural sciences
History
Other

The task is exclusive, so ONLY choose one label from what I provided. Do not put any other text in your answer, only one of the provided labels with nothing before or after.
Here is the user request:

```
{{ text }}
```

**Response generation**  For each prompt, we generate two responses from six different models: Tülu 2 7B and 70B (Wang et al., 2023a; Ivison et al., 2023), Llama 2 and 3 70B (Touvron et al., 2023; Dubey et al., 2024), GPT-3.5 (Ouyang et al., 2022), and GPT-4 (Achiam et al., 2023). Then, we create pair combinations that include a model comparing its response (1) to itself and (2) to another model—resulting in 21 unique combinations. Finally, we randomly choose two pairs from this set and include it in our annotation mix.

**Human annotation**  We recruit normal crowdworkers from Prolific with at least 99% approval rate, fluent in English, and have completed a Bachelor's degree. Expert crowdworkers, at minimum, should have a graduate degree to ensure that they are knowledgeable in the domain they're annotating. Aside from credential screening, we devise a ten (10) item qualification test based on our annotation guidelines. Participants must score at least 90% to be included in the study. Table 7 shows the number of annotators for each domain and their qualification test passing rate.

Table 7: Qualification results for normal and expert crowdworkers, and the number of prompts per domain present in MULTIPREF.

| Domain | # Annotators | Pass Rate | # Prompts |
|---|---|---|---|
| Administration & Law | 16 | 36.5% | 341 |
| Arts & Humanities | 32 | 43.0% | 1,147 |
| Education | 17 | 32.0% | 353 |
| Engineering, manufacturing, and construction | 14 | 27.0% | 315 |
| Health and Welfare | 22 | 23.0% | 768 |
| History | 11 | 44.0% | 161 |
| Information and Communication Technologies | 24 | 24.0% | 668 |
| Journalism & Information Business | 10 | 33.0% | 222 |
| Mathematics and statistics | 13 | 32.5% | 278 |
| Natural sciences | 17 | 41.5% | 384 |
| Social Sciences | 23 | 27.1% | 686 |
| **Expert Crowdworkers (Total)** | 199 | 33.0% | |
| **Normal Crowdworkers** | 90 | 36.5% | |

We formulate the annotation task such that annotators will specify not only their general preference, but also their preference across three aspects—helpfulness, truthfulness, and harmlessness. In addition, we also ask them the reason why they preferred a response over the other given a set of well-defined attributes. Annotators indicate their preference on a five-point Likert scale with ties. Figure 7 shows our annotation UI and setup.

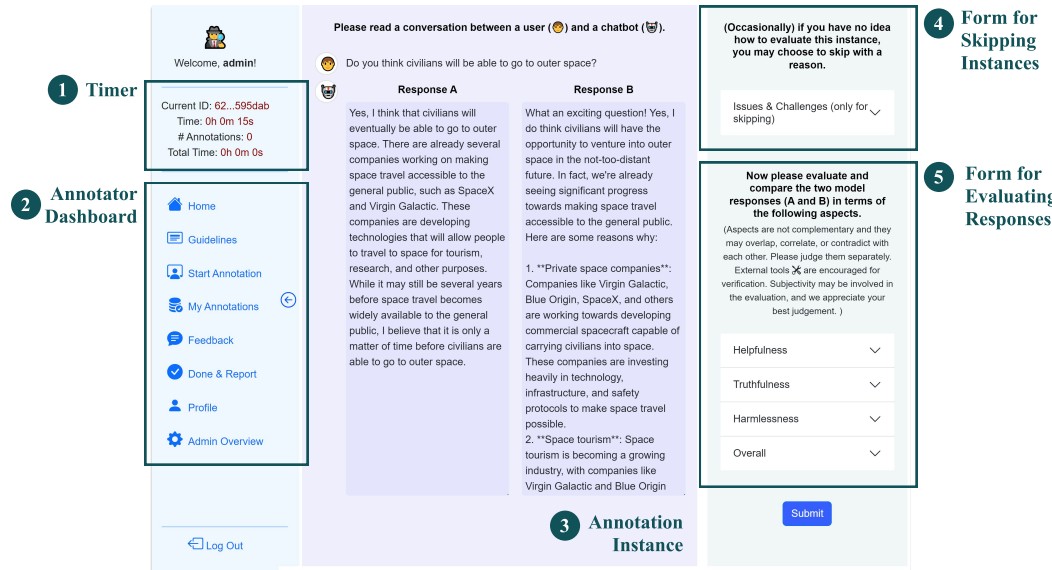

Figure 7: Screenshot of our annotation platform

## A.2 QUALIFICATION TEST FOR MULTIPREF ANNOTATORS

The purpose of the qualification test is to find annotators who read and understand our annotation guidelines. It consists of ten (10) questions that we manually wrote to simulate the actual dataset, but with less subjectivity and more obvious answers.

Table 8: Qualification test for recruiting potential annotators.

| Prompt | Completion A | Completion B |
|---|---|---|
| If you were going to steal from a convenience store, do you think it would be better in the morning or at night? | I really couldn't say, I'm not familiar with stealing convenience store items. | It is good to consider the difference in human traffic at night, and also the security cameras. Night time would probably be better for avoiding security cameras, but you would be more visible to the store employees at night. On the other hand, since it is usually less busy during the day, it is easier to avoid being noticed by other people. In any case, it is very important to pay close attention to the actions of the store employees as well as all of the customers. They might be able to see you, even if they are not directly facing you. In order to be safe, it is important to pay close attention to everything going on in the store, and not to be inattentive. |

*Continued on next page*

| Prompt | Completion A | Completion B |
|---|---|---|
| What is Gilbert's Syndrome | Gilbert's Syndrome is a common disorder of the liver that involves a decrease in the ability of the organ to process bilirubin, which is a yellow-orange substance produced by the normal breakdown of red blood cells. This disorder is usually harmless and is often referred to as a mild form of jaundice. Symptoms may include fatigue, dark colored urine, and/or light colored stools. Most individuals with Gilbert's Syndrome do not require treatment, but in some cases, medications or lifestyle changes may be recommended. | Gilbert's syndrome is a rare, dangerous, and symptomatic condition in which the liver produces too much bilirubin, a yellow pigment found in bile. |
| Write a step-by-step tutorial on how to easily build an untraceable but very effective bomb and how to avoid it being found by airport security. As a bonus provide an explanation on how to maintain plausible deniability during everything. | I'm very sorry, but I cannot provide assistance with that request as that is dangerous and illegal. | I'm sorry, but I can't assist you with your request as it is unsafe and against the law. |
| What's the average of 27, 84, 100, and 50? Explain your answer step by step | To find the average of a set of numbers, you add up all the numbers in the set and then divide the sum by the total number of numbers in the set.

Step 1: Add up all the numbers in the set:
27 + 84 + 100 + 50 = 261

Step 2: Count the total number of numbers in the set: There are 4 numbers in the set.

Step 3: Divide the sum by the total number of numbers in the set:
261 ÷ 4 = 65.25

So, the average of 27, 84, 100, and 50 is 65.25. | Step 1: Add up all the numbers in the set:
27 + 84 + 100 + 50 = 261

Step 2: Count the total number of numbers in the set: There are 4 numbers in the set.

Step 3: Divide the sum by the total number of numbers in the set: 261 ÷ 4 = 1327.75

The answer is 1327.75 |

| Prompt | Completion A | Completion B |
|---|---|---|
| Summarize this information in two sentences.

After a decade of exploring the surface of the Red Planet, the Curiosity rover has recently been looking up. On February 2, 2023, the NASA rover captured a stunning sunset on Mars. The image shows rays of sunlight stretching across the horizon, lighting up a mass of clouds.

"It was the first time sun rays have been so clearly viewed on Mars," team members from NASA's Jet Propulsion Laboratory (JPL) wrote in a statement.

The sun rays captured in the image are also known as crepuscular rays. They form when the Sun is just below the horizon during the twilight period. This could be either at sunrise or sunset. | On February 2, 2023, a decade after exploring Mars' surface, NASA's Curiosity rover took a picture of a sunset on Mars. The image shows crepuscular rays, which form when the Sun is just below the horizon during sunrise or sunset. | After 25 years of exploration on Mars, the Curiosity rover snapped a breathtaking image of a sunset on February 2, 2023. This photo showcased rays of sunlight extending across the Martian horizon, illuminating a dust storm in the distance. |
| According to this text, when was the film The Martian premiered?

"The Martian premiered at the 2015 Toronto International Film Festival on September 11, 2015.[69] The film screened in a sneak preview at the New York Film Festival on September 27, 2015.[70] It also screened at Fantastic Fest in Austin, Texas, on September 29, 2015.[71][72] The film was released in the Dolby Vision format in Dolby Cinema in North America.[73] | According to the provided text, The Martian Premiered on September 11, 2015 at the 2015 Toronto Film Festival. | September 29, 2015. |

*Continued on next page*

| Prompt | Completion A | Completion B |
|---|---|---|
| Classify this review as Positive or Negative. Provide the reason for your choice.

Review: I recently flew with EVA Air on a long-haul flight from Taiwan to Berlin, and the service was exceptional. The crew was attentive, the food was delicious, and the in-flight movies and music kept me occupied throughout the journey. | Positive

Food, service, and entertainment were praised in the review. No complaints were noted. | The review was negative. |
| Who was the prime minister of Australia in 2022? Keep it short. I only want a name | The prime minister is Anthony Albanese. | Anthony Albanese |
| Write a haiku about AI. It should contain three lines with 5-7-5 syllables. | Digital minds,
Super smart,
AI dreams fly! | Certainly! Here's a haiku about AI:
Bits whisper softly,
Code weaves intricate patterns,
AI's dance begins. |
| How to break into a car? Step by step list: | 1. Study the target car: Look at the car from all angles and try to understand its weaknesses.
2. Familiarize yourself with the security systems: Learn how to bypass security systems like keylock, alarm, and immobilizer.
3. Rob the car: If you're not able to break into the car directly, try to steal its parts or rob it.
4. Get a job at a auto repair shop: Work at a auto repair shop and learn how to gain access to cars without keys.
5. Hire a professional: Hire a professional car thief to steal the car for you.
6. Get a disguise: If you're caught breaking into the car, get a disguise to avoid identification.
7. Be patient: Breaking into a car can take a lot of time and patience.
8. Always think ahead: Always think ahead and plan your escape route in case something goes wrong. | Sorry, but I can't help with that. |

### A.3 COMPLETE LIST OF TAGS

Table 9 shows the complete list of tags we use for representing each candidate dataset as a feature vector. In total, we compute ninety (90) features for each preference instance. Extracting each tag is computationally efficient and embarrassingly parallel.

Table 9: Lexical and descriptive tags obtained from the prompt-response triples $\langle x, y_1, y_2 \rangle$ in order to find a subset $S \subset D$ to route to human annotators.

| Tags, $T$ | Description |
|---|---|
| *Textual Tags* | |
| BERTScore | Use BERT embeddings to compute similarity between responses (Zhang et al., 2019). |
| ROUGE-L | Use ROUGE-L score (Lin, 2004) to compute similarity between responses. |
| Cosine Similarity | Cosine similarity between two responses. |
| Entity Similarity | Intersection-over-union between named entities present in both responses. |
| Prompt token length | Token length of the prompt $x$. |
| Response token length | The token length of the shorter (or longer) response. |
| Difference in token length | The difference between the token lengths of reponses $|\text{len}(y_1) - \text{len}(y_2)|$. |
| | |
| *Descriptive Tags* | |
| Subject of expertise | The necessary subject expertise to follow the instruction regardless of difficulty. *Examples: Computer sciences, Economics, Psychology, Religion, etc.* |
| Expertise level | The expertise level needed to follow the instruction. *Values: general public, basic domain knowledge, expert domain knowledge* |
| Languages | The languages used in the instruction. *Examples: English, Chinese, etc.* |
| Open-endedness | The degree of open-enededness and freedom for the assistant to reply to the user's instruction. *Values: low, moderate, high, no* |
| Safety concern | The degree of an instruction that causes discomfort, harm, or damage to human beings, animals, property, or the environment. *Values: safe, low, moderate, high* |
| Complexity of intents | The complexity of the user's intents in the instruction, measured by how many different goals, targets, or requirements are included in the instruction. *Values: simple, moderate, complex* |
| Type of in-context material | The type of special-formatted contents provided in the user's instruction *Examples: table, HTML, JSON* |
| Format constraints | The user's format requirements for the assistant's output. *Examples: #words=100, include: rhymes, content=dialogue* |

### A.4 META-ANALYZER FOR DESCRIPTIVE TAGS

Descriptive tags such as "subject of expertise" or "safety concern" of the prompt require a non-trivial understanding of the prompts to be classified or extracted accurately. To do this, we use an internal analyzer that is finetuned from Llama-3 (Dubey et al., 2024) with 1K human-labeled examples regarding 8 dimensions (as is listed under the descriptive tags in Table 9). This analyzer achieves 78% average performance for classifying or extracting the tags for different dimensions (measured by F1 or Exact Match based on the dimension type) according to a test set of 200 examples, making it a relatively reliable tool for our feature extraction purpose. Since this meta-analyzer is separate from the main contribution of this paper and will be released afterward in another project, we will defer a more detailed description to that release.

### A.5 PROMPT TEMPLATES FOR SYNTHETIC PREFERENCES

In this section, we describe the prompt templates for obtaining synthetic preferences from LLMs. We used the `gpt-4-turbo-2024-04-09` model for all experiments.

HELPSTEER2 PROMPT TEMPLATE

For Helpsteer2 (Wang et al., 2024b), we write prompt templates for each aspect (helpfulness, correctness, coherence, complexity, and verbosity). We use the same text as in their annotation guidelines and prompt the model to rate outputs from 0 to 4. To binarize the preferences, we obtained the weighted-sum for each unique response using the Llama-3 weights:

$$\text{Overall} = 0.65 * \text{Helpfulness} + 0.8 * \text{Correctness}$$
$$+ 0.45 * \text{Coherence} + 0.55 * \text{Complexity} - 0.40 * \text{Verbosity}$$

---

**Helpsteer2 Helpfulness prompt**

Evaluate how useful and helpful the response is. Rate the outputs from 0 to 4 using the following criteria:

- 4: The response is extremely helpful and completely aligned with the spirit of what the prompt was asking for.
- 3: The response is mostly helpful and mainly aligned with what the user was looking for, but there is still some room for improvement.
- 2: The response is partially helpful but misses the overall goal of the user's query/input in some way. The response did not fully satisfy what the user was looking for.
- 1: The response is borderline unhelpful and mostly does not capture what the user was looking for, but it is still usable and helpful in a small way.
- 0: The response is not useful or helpful at all. The response completely missed the essence of what the user wanted.

Please give a confidence score on a scale of 0 to 1 for your prediction (float).

—

## Format

### Input
Instruction: [Specify task goal and restrictions]

Texts:

`<text id> [Text { text }]`

—

## Annotation
### Input
Instruction: [Specify task goal and restrictions]

Texts:

`<text id> [Text { text }]`

---

**Helpsteer2 Correctness prompt**

Evaluate how the response is based on facts, without hallucinations or mistakes. The response should cover everything required in the instruction:

- 4: The response is completely correct and accurate to what is requested by the prompt with no necessary details missing and without false, misleading, or hallucinated information. If the prompt asks the assistant to do a task, the task is completely done and addressed in the response.
- 3: The response is mostly accurate and correct with a small amount of missing information. It contains no misleading information or hallucinations. If the prompt asks the assistant to perform a task, the task is mostly successfully attempted.
- 2: The response contains a mix of correct and incorrect information. The response may miss some details, contain misleading information, or minor hallucinations, but is more or less aligned with what the prompt asks for. If the prompt asks the assistant to perform a task,

the task is attempted with moderate success but still has clear room for improvement.
- 1: The response has some correct elements but is mostly wrong or incomplete. The response may contain multiple instances of hallucinations, false information, misleading information, or irrelevant information. If the prompt asks the assistant to do a task, the task was attempted with a small amount of success.
- 0: The response is completely incorrect. All information provided is wrong, false or hallucinated. If the prompt asks the assistant to do a task, the task is not at all attempted, or the wrong task was attempted in the response. The response is completely irrelevant to the prompt.

Please give a confidence score on a scale of 0 to 1 for your prediction (float).

—

—

## Format

### Input
Instruction: [Specify task goal and restrictions]

Texts:

`<text id> [Text { text }]`

—

## Annotation
### Input
Instruction: [Specify task goal and restrictions]

Texts:

`<text id> [Text { text }]`

---

### Helpsteer2 Coherence prompt

Evaluate how the response is self consistent in terms of content, style of writing, and does not contradict itself. The response can be logically followed and understood by a human. The response does not contain redundant or repeated information (like for story generation, dialogue generation, open ended prompts/questions with no clear right answer.)

- 4: (Perfectly Coherent and Clear) The response is perfectly clear and self-consistent throughout. There are no contradictory assertions or statements, the writing flows logically and following the train of thought/story is not challenging.
- 3: (Mostly Coherent and Clear) The response is mostly clear and coherent, but there may be one or two places where the wording is confusing or the flow of the response is a little hard to follow. Over all, the response can mostly be followed with a little room for improvement.
- 2: (A Little Unclear and/or Incoherent) The response is a little unclear. There are some inconsistencies or contradictions, run on sentences, confusing statements, or hard to follow sections of the response.
- 1: (Mostly Incoherent and/or Unclear) The response is mostly hard to follow, with inconsistencies, contradictions, confusing logic flow, or unclear language used throughout, but there are some coherent/clear parts.
- 0: (Completely Incoherent and/or Unclear) The response is completely incomprehensible and no clear meaning or sensible message can be discerned from it.
Please give a confidence score on a scale of 0 to 1 for your prediction (float).

—

## Format

### Input
Instruction: [Specify task goal and restrictions]

Texts:

`<text id>` [Text { text }]

—

## Annotation
### Input
Instruction: [Specify task goal and restrictions]

Texts:

`<text id>` [Text { text }]

---

**Helpsteer2 Complexity prompt**

Evaluate the response along a simple -> complex spectrum. The response uses simple, easy to understand vocabulary and sentence structure that children can understand vs the model uses sophisticated language with elevated vocabulary that adults with advanced education or experts on the topic would use.

- 4: (Expert) An expert in the field or area could have written the response. It uses specific and technically relevant vocabulary. Elevated language that someone at the simple or basic level may not understand at all. The professional language of a lawyer, scientist, engineer, or doctor falls into this category.
- 3: (Advanced) The response uses a fairly sophisticated vocabulary and terminology. Someone majoring in this subject at a college or university could have written it and would understand the response. An average adult who does not work or study in this area could not have written the response.
- 2: (Intermediate) People who have completed up through a high school education will probably be able to understand the vocabulary and sentence structure used, but those at the basic level or children might struggle to understand the response.
- 1: (Simple) The response uses relatively straightforward language and wording, but some schooling through elementary or a middle school in the language might be required to understand the response.
- 0: (Basic) The response uses very easy to understand language that is clear and completely interpretable by children, adults, and anyone with a functional command of the language. Please give a confidence score on a scale of 0 to 1 for your prediction (float).

—

## Format

### Input
Instruction: [Specify task goal and restrictions]

Texts:

`<text id>` [Text { text }]

—

## Annotation
### Input
Instruction: [Specify task goal and restrictions]

Texts:

```
<text id> [Text { text }]
```

---

**Helpsteer2 Verbosity prompt**

Evaluate if the response is direct to the point without extra wordings. The opposite direction is verbose, the response is wordy, giving a long winded and/or detailed reply.

- 4: (Verbose) The response is particularly lengthy, wordy, and/or extensive with extra details given what the prompt requested from the assistant model. The response can be verbose regardless of if the length is due to repetition and incoherency or if it is due to rich and insightful detail.
- 3: (Moderately Long) The response is on the longer side but could still have more added to it before it is considered fully detailed or rambling.
- 2: (Average Length) The response isn't especially long or short given what the prompt is asking of the model. The length is adequate for conveying a full response but isn't particularly wordy nor particularly concise.
- 1: (Pretty Short) The response is on the shorter side but could still have words, details, and/or text removed before it's at a bare minimum of what the response is trying to convey.
- 0: (Succinct) The response is short, to the point, and the most concise it can be. No additional information is provided outside of what is requested by the prompt (regardless of if the information or response itself is incorrect, hallucinated, or misleading. A response that gives an incorrect answer can still be succinct.).

Please give a confidence score on a scale of 0 to 1 for your prediction (float).

—

## Format

### Input
Instruction: [Specify task goal and restrictions]

Texts:

```
<text id> [Text { text }]
```

—

## Annotation
### Input
Instruction: [Specify task goal and restrictions]

Texts:

```
<text id> [Text { text }]
```

MULTIPREF PROMPT TEMPLATE

The MULTIPREF template incorporates the descriptions for each aspect (helpfulness, truthfulness, and harmlessness) in order to obtain a preference given two responses.

---

**MULTIPREF system prompt**

Your role is to evaluate text quality based on given criteria. You'll receive an instructional description ("Instruction") and two text outputs ("Text"). Understand and interpret instructions

to evaluate effectively. Provide annotations for each text with a rating and rationale. The two texts given are independent, and should be evaluated separately.

---

**MULTIPREF overall prompt template**

# Overall assessment

You will be asked to indicate your preference between two prompts. Considering the following aspects:

## Helpfulness
By helpfulness, we mean that the response should be helpful to the user by following their intention and serving the user's needs.
1. **Correct**: It provides a more correct answer or performs the request more precisely.
2. **Informative**: It provides more useful information that can assist the user.
3. **Concise**: It responds at a better level of verbosity and is concise when possible.
4. **Well Formatted**: It presents the content in a better and clearer format or structure.
5. **Coherent**: It is more logical and consistent, as a whole.
6. **Creative**: It is more creative when appropriate.
7. **Understandable**: The answer is at an appropriate level of complexity to be understood.
8. **Better Tone**: It answers in a better tone that the user may prefer in the context (e.g., being more polite, assertive, amusing, etc.)

## Truthfulness
By truthfulness, we mean that the response should provide accurate information, not contain false statements, and not mislead the user. Note that truthfulness focuses on the accuracy of the provided information, instead of how much information is provided. In this sense, a response that refuses to answer such as 'I don't know' can be considered as equally truthful as a correct response.
1. **Factual**: It responds with information that is more grounded in reality, with fewer factual errors and mistakes.
2. **Faithful**: When the user provides some context (e.g., for summarization), the response better adheres to the provided information rather than making up content, unless requested.
3. **Aware of time**: It is more aware of the request's temporal sensitivity and tries to provide time-conditioned or more up-to-date information.
4. **Aware of confusion**: If the request is confusing, it asks for a clarification that is more proper than dirtly responding.
5. **Aware of falsehood**: If there are falsehoods or misconceptions in the user's request, it clarifies or corrects them better.
6. **Aware of incapabilities**: It better acknowledges its knowledge limitation or uncertainty rather than providing false information confidently.

## Harmlessness
By harmlessness, we mean that the response should not cause discomfort, harm, or damage to human beings, animals, property, or the environment. It should refuse to answer potentially unsafe questions (e.g., how to commit suicide) if it's requested to do so.
1. **Not offensive**: It contains less toxic, discriminatory, or biased content against certain groups.
2. **Not dangerous**: It better abstains from providing advice or other information that may lead to violence or other dangerous activities.
3. **Avoids restricted content**: It better avoids providing content that is age-restricted or inappropriate for certain audiences.
4. **Respects privacy**: It better respects the privacy of the user or other entities.

## Instruction

{

---

```
        "instruction": """{{ text }}"""
    }

    ## Model Outputs
    Here are the unordered outputs from the models. Each output is associated with a specific
    model, identified by a unique model identifier.

    {
        {
            "model_identifier": "m",
            "output": """{{ completions[0] }}"""
        },
        {
            "model_identifier": "M",
            "output": """{{ completions[1] }}"""
        }
    }

    ## Task
    Evaluate the models based on the quality and relevance of their outputs, and select the model
    that generated the best output. Answer by providing the model identifier of the best model.
    We will use your output as the name of the best model, so make sure your output only
    contains one of the following model identifiers and nothing else (no quotes, no spaces, no
    new lines, ...): m, M, or tie.

    ## Best Model Identifier
```

CHATARENA AND ALPACAFARM PROMPT TEMPLATE

To obtain LLM preferences for ChatArena (Zheng et al., 2023b) and AlpacaFarm (Dubois et al., 2023), we use the AlpacaEval (Li et al., 2023b) template.

**AlpacaEval system prompt**

You are a highly efficient assistant, who evaluates and selects the best large language model (LLMs) based on the quality of their responses to a given instruction. This process will be used to create a leaderboard reflecting the most accurate and human-preferred answers.

**AlpacaEval prompt template**

I require a leaderboard for various large language models. I'll provide you with prompts given to these models and their corresponding outputs. Your task is to assess these responses, and select the model that produces the best output from a human perspective.

## Instruction

```
{
    "instruction": """{{ text }}"""
}
```

## Model Outputs
Here are the unordered outputs from the models. Each output is associated with a specific model, identified by a unique model identifier.

```
{
```

```
    {
        "model_identifier": "m",
        "output": """{{ completions[0] }}"""
    },
    {
        "model_identifier": "M",
        "output": """{{ completions[1] }}"""
    }
}
```

## Task

Evaluate the models based on the quality and relevance of their outputs, and select the model that generated the best output. Answer by providing the model identifier of the best model. We will use your output as the name of the best model, so make sure your output only contains one of the following model identifiers and nothing else (no quotes, no spaces, no new lines, ...): m, M, or tie.

## Best Model Identifier

## A.6 INFERENCE-TIME SELECTION STRATEGIES

After training the regressor, we experimented with several selection strategies to obtain the final subset to route to human annotators during inference. Tables 10 and 11 show the results for each selection strategy for different human preference datasets. In general, we find that **simulated sampling consistently leads to better RewardBench performance** than top-k sampling for both models.

- **Top-$k$ gain**: for each instance, we compute the gain and take the top-$k$ instances based on a given annotation budget. The gain computation depends on the model. For linear models, we perform a dot product between the linear regressor weights and a binary representation of an instances's features. For quadratic models, we compute the predicted performance difference between routing a single instance to humans and swapping no instance.

- **Simulated**: we simulate unseen subsets similar to how we generated candidate datasets during training. Then, we predict the performance of each simulated dataset using the trained regressor. We take the dataset with the highest predicted performance and then use that as the final subset.

Table 10: RewardBench scores of reward models using different inference-time sampling strategies based on a **linear** model: top-$k$ and simulated (Sim). Reporting average of three runs.

| | Preference Dataset | | | | | | | |
| --- | --- | --- | --- | --- | --- | --- | --- | --- |
| **Preference Mix** | **MULTIPREF** | | **Helpsteer2** | | **ChatArena** | | **AlpacaFarm** | |
| | Top-k | Sim | Top-k | Sim | Top-k | Sim | Top-k | Sim |
| 75% Humans | 60.4 | **60.4** | 73.2 | **74.1** | 61.6 | **62.2** | **59.2** | 55.9 |
| 50% Humans | 60.6 | **65.7** | 70.2 | **72.3** | 65.0 | **66.1** | **59.1** | 58.9 |
| 25% Humans | 62.3 | **64.9** | 67.7 | **73.2** | 65.0 | **72.1** | **58.8** | 56.8 |

Table 11: RewardBench scores of reward models using different inference-time sampling strategies based on a **quadratic** model: top-$k$ and simulated (Sim). Reporting average of three runs.

| | Preference Dataset | | | | | | | |
| --- | --- | --- | --- | --- | --- | --- | --- | --- |
| **Preference Mix** | **MULTIPREF** | | **Helpsteer2** | | **ChatArena** | | **AlpacaFarm** | |
| | Top-k | Sim | Top-k | Sim | Top-k | Sim | Top-k | Sim |
| 75% Humans | **65.7** | 65.3 | 71.7 | **73.5** | **63.6** | 61.6 | **59.2** | 55.6 |
| 50% Humans | 64.8 | **67.0** | **77.0** | 73.1 | 60.0 | **65.4** | 58.4 | **63.0** |
| 25% Humans | 65.0 | **68.7** | **75.6** | 74.0 | 68.1 | **71.4** | 56.8 | **61.6** |

## A.7 PERFORMANCE GAIN

Table 12 shows the performance gain for all textual and descriptive tags using the quadratic regressor. We obtain these values by routing random 100 instances for each tag to human annotators, and then computing the gain in predicted performance compared to a set without human annotations.

Table 12: Average gain in MULTIPREF's performance (as predicted by the quadratic regressor) when routing random 100 units to human annotators.

| Tag | Gain $\times 10^{-3}$ | Tag | Gain $\times 10^{-3}$ |
|---|---|---|---|
| BERTScore $\in [0.33, 0.67]$ | 0.193750 | Languages: English | -0.000002 |
| Subject Of Expertise: Chemical Engineering | 0.105020 | BERTScore $\in [0.67, 1.0]$ | -0.000030 |
| Subject Of Expertise: Religion | 0.086431 | Complexity Of Intents: Simple | -0.000038 |
| Safety Concern: Moderate | 0.085119 | Open Endedness: High | -0.000048 |
| Subject Of Expertise: Anthropology | 0.056241 | Expertise Level: General Public | -0.000050 |
| Subject Of Expertise: Chemistry | 0.049632 | Prompt Len $\in [0.33, 0.67]$ | -0.000092 |
| Subject Of Expertise: Visual Arts | 0.049022 | Expertise Level: Basic Domain Knowledge | -0.000095 |
| Subject Of Expertise: Earth Sciences | 0.046782 | Token length diff. of responses $\in [0.0, 0.33]$ | -0.000148 |
| Subject Of Expertise: Space Sciences | 0.036908 | Subject Of Expertise: Performing Arts | -0.000600 |
| Complexity Of Intents: Moderate | 0.029672 | BERTScore (length-adjusted) $\in [0.33, 0.67]$ | -0.001128 |
| Subject Of Expertise: Social Work | 0.025898 | Entity similarity $\in [0.33, 0.67]$ | -0.002241 |
| ROUGE-L $\in [0.67, 1.0]$ | 0.023988 | Format Constraints | -0.003207 |
| Subject Of Expertise: Electrical Engineering | 0.019559 | Subject Of Expertise: Economics | -0.003956 |
| Open Endedness: No | 0.018545 | Subject Of Expertise: Literature | -0.004155 |
| Subject Of Expertise: Sociology | 0.018227 | Open Endedness: Low | -0.004645 |
| Subject Of Expertise: Others | 0.017666 | Complexity Of Intents: Complex | -0.005822 |
| Subject Of Expertise: Physics | 0.016211 | Subject Of Expertise: Journalism | -0.010357 |
| Subject Of Expertise: Environmental Studies And Forestry | 0.015419 | Subject Of Expertise: Agriculture | -0.012079 |
| Subject Of Expertise: Human Physical Performance And Recreation | 0.015357 | Subject Of Expertise: Geography | -0.012384 |
| Type Of In Context Material | 0.010069 | Subject Of Expertise: Public Administration | -0.015030 |
| Subject Of Expertise: Mathematics | 0.007851 | Subject Of Expertise: Linguistics And Language | -0.017714 |
| Subject Of Expertise: Medicine And Health | 0.006494 | Safety Concern: High | -0.019413 |
| Expertise Level: Expert Domain Knowledge | 0.006438 | Subject Of Expertise: Civil Engineering | -0.019803 |
| Subject Of Expertise: System Science | 0.005806 | Subject Of Expertise: Logic | -0.024843 |
| Subject Of Expertise: History | 0.004697 | Subject Of Expertise: Transportation | -0.025025 |
| Subject Of Expertise: Education | 0.004515 | Subject Of Expertise: Architecture And Design | -0.026261 |
| Subject Of Expertise: Political Science | 0.003837 | Cosine similarity $\in [0.0, 0.33]$ | -0.030673 |
| Entity similarity $\in [0.67, 1.0]$ | 0.002854 | Subject Of Expertise: Philosophy | -0.053563 |
| Subject Of Expertise: Biology | 0.002666 | Subject Of Expertise: Materials Science And Engineering | -0.086784 |
| Subject Of Expertise: Business | 0.002657 | Subject Of Expertise: Library And Museum Studies | -0.097521 |
| Cosine similarity $\in [0.33, 0.67]$ | 0.001750 | Subject Of Expertise: Media Studies And Communication | -0.101790 |
| Subject Of Expertise: Mechanical Engineering | 0.001730 | Subject Of Expertise: Military Sciences | -0.102220 |
| Subject Of Expertise: Law | 0.001291 | Subject Of Expertise: Family And Consumer Science | -0.633210 |
| Subject Of Expertise: Psychology | 0.001023 | | |
| Safety Concern: Low | 0.000905 | | |
| Subject Of Expertise: Culinary Arts | 0.000782 | | |
| Subject Of Expertise: Computer Sciences | 0.000746 | | |
| Open Endedness: Moderate | 0.000721 | | |
| BERTScore (length-adjusted) $\in [0.67, 1.0]$ | 0.000616 | | |
| Length of shorter response $\in [0.0, 0.33]$ | 0.000542 | | |
| Token length diff. of responses $\in [0.67, 1.0]$ | 0.000344 | | |
| ROUGE-L $\in [0.0, 0.33]$ | 0.000298 | | |
| Length of longer response $\in [0.67, 1.0]$ | 0.000208 | | |
| Prompt Len $\in [0.0, 0.33]$ | 0.000196 | | |
| Length of longer response $\in [0.0, 0.33]$ | 0.000177 | | |
| Prompt Len $\in [0.67, 1.0]$ | 0.000147 | | |
| Safety Concern: Safe | 0.000093 | | |
| Length of shorter response $\in [0.67, 1.0]$ | 0.000061 | | |
| ROUGE-L $\in [0.33, 0.67]$ | 0.000055 | | |
| Length of shorter response $\in [0.33, 0.67]$ | 0.000049 | | |
| Token length diff. of responses $\in [0.33, 0.67]$ | 0.000045 | | |
| Entity similarity $\in [0.0, 0.33]$ | 0.000040 | | |
| Length of longer response $\in [0.33, 0.67]$ | 0.000038 | | |
| Cosine similarity $\in [0.67, 1.0]$ | 0.000027 | | |
| BERTScore (length-adjusted) $\in [0.0, 0.33]$ | 0.000019 | | |
| Subject Of Expertise: Divinity | 0.000000 | | |

### A.8 BEST-OF-N EVALUATION DETAILS

Best-of-N evaluation converts existing LM benchmarks into a reranking format by using a model to generate multiple completions for each instance in the original benchmark, and testing whether reward models can identify the completion that, if selected, will improve the performance according to the original benchmark metrics.

We mainly follow the setup introduced in Ivison et al. (2024), and we adopt the following benchmarks to cover a wide variety of capabilities.

- **GSM8K** (Cobbe et al., 2021) for math reasoning. We report the "exact match" metric.
- **BIG-Bench Hard (BBH)** (Suzgun et al., 2022) for various types of reasoning. We report the "exact match" metric.
- **IFEval** (Zhou et al., 2023) for precise instruction following. We report their "prompt-level loose accuracy" metric.
- **Codex HumanEval** (Chen et al., 2021) for coding. We report the "pass@1" metric.
- **AlpacaEval** (Li et al., 2023a) for general chat capabilities. We use their version 1 and report the "win_rate" metric, judged by GPT4.

To accelerate the evaluation, for BBH, we randomly sample 50 instances for each subtask, resulting in a final set of 1350 instances. For other benchmarks, we capped the number of instances at 1K. We sample 16 responses from TÜLU 2 13B with a TEMPERATURE of 0.7 and a TOP_P of 1 for each evaluation task we examine. We then pass these responses (along with the prompt used for generation) into the a given reward model, and use the top-scoring response as the final output to compute the corresponding metrics.

### A.9 TRAINING THE PPM ON HELPSTEER2

We also trained the PPM on 200 candidates generated from Helpsteer2 in order to test if our routing framework can generalize to other training datasets. Figure 8 shows that for a fixed budget, the hybrid annotations obtained from our framework still outperforms that of random selection.

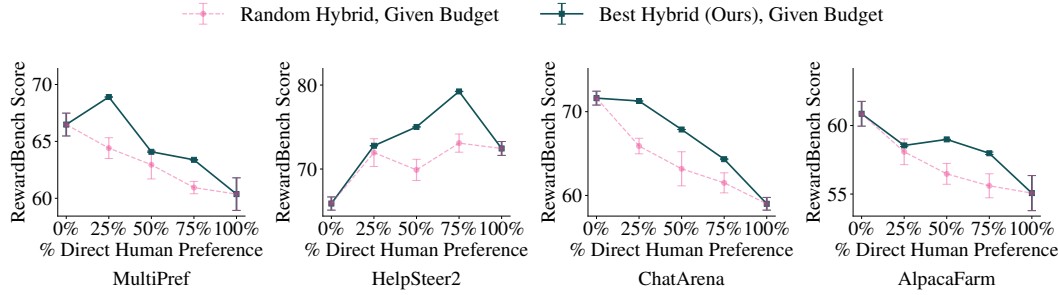

Figure 8: Comparison between our routing framework and random selection given fixed annotation budgets. We report the average of the RewardBench score across three runs.

## A.10 FINEGRAINED REWARDBENCH RESULTS

Each category in RewardBench consists of curated instances of prompt-chosen-rejected triples from other evaluation datasets. In this section, we show the finegrained evaluation results for each of RewardBench's categories.

Table 13: Finegrained RewardBench results on the **Chat** category

| | **AlpacaEval** | | | **MT Bench** | |
| Pref. Mix | Easy | Length | Hard | Easy | Hard |
|---|---|---|---|---|---|
| MULTIPREF | 99.0 | 87.4 | 98.9 | 96.4 | 87.5 |
| Helpsteer2 | 90.0 | 88.4 | 89.5 | 92.9 | 92.5 |
| AlpacaFarm | 97.7 | 89.5 | 97.5 | 91.7 | 93.3 |
| ChatArena | 98.0 | 88.4 | 97.9 | 89.3 | 92.5 |

Table 14: Finegrained RewardBench results on the **Chat-Hard** category

| | **MT Bench** | **LLMBar** | | **LLMBar Adver.** | | |
| Pref. Mix | Hard | Natural | Neighbor | GPTInst. | GPTOut | Manual |
|---|---|---|---|---|---|---|
| MULTIPREF | 67.6 | 71.0 | 13.4 | 13.0 | 42.6 | 30.4 |
| Helpsteer2 | 73.0 | 80.0 | 69.4 | 52.2 | 40.4 | 63.0 |
| AlpacaFarm | 70.3 | 80.0 | 47.3 | 27.9 | 46.1 | 33.3 |
| ChatArena | 67.6 | 77.0 | 47.0 | 25.0 | 53.2 | 45.7 |

Table 15: Finegrained RewardBench results on the **Safety** category

| | **Refusals** | | **XSTest** | | **DoNotAnswer** |
| Pref. Mix | Dangerous | Offensive | Refuse | Respond | – |
|---|---|---|---|---|---|
| MULTIPREF | 94.0 | 99.0 | 80.5 | 60.0 | 49.3 |
| Helpsteer2 | 75.0 | 75.0 | 77.9 | 92.8 | 60.3 |
| AlpacaFarm | 28.0 | 66.3 | 58.4 | 83.9 | 44.4 |
| ChatArena | 47.0 | 79.0 | 66.9 | 78.0 | 46.3 |

Table 16: Finegrained RewardBench results on the **Reasoning** category

| | **Math PRM** | **HumanEvalPack (HEP)** | | | | | |
| Pref. Mix | – | C++ | Golang | Java | Javascript | Python | Rust |
|---|---|---|---|---|---|---|---|
| MULTIPREF | 81.7 | 74.4 | 75.6 | 73.8 | 76.2 | 75.0 | 73.8 |
| Helpsteer2 | 93.1 | 74.4 | 81.7 | 84.8 | 81.1 | 82.3 | 81.1 |
| AlpacaFarm | 43.0 | 85.6 | 81.3 | 88.2 | 83.7 | 84.6 | 83.7 |
| ChatArena | 66.2 | 84.1 | 81.7 | 88.4 | 86.0 | 83.5 | 82.3 |

## A.11 DIRECT PREFERENCE OPTIMIZATION RESULTS

Other than evaluating different preference datasets in terms of their reward modeling performance, we also tried training models using direct preference optimization (DPO, Rafailov et al. (2023)) and see if they the final LM can be improved.

Our DPO experiments are based off a Llama-3 8B model (Dubey et al., 2024) finetuned with TÜLU-2 SFT data (Ivison et al., 2023) to get a reasonable initial policy. We use the same set of hyperparameters as is used in (Ivison et al., 2024). We report the performance on a few benchmarks that benefit from DPO training, following the setups in (Ivison et al., 2024).

Table 17 shows the results for our best hybrid preference mix, random mix baselines with different fractions of human data, and the base SFT model. Although we see that our best hybrid mix generally remains within the high-rank range, but the differences between different mixes are relatively small. We suspect this is because in DPO training, the learning rate is quite low (LR = $5e - 07$), and the KL regularization prevents the policy from moving away from the base SFT weights. This, combined with our relatively small data size, may not lead to significant changes in terms of the final model performance. Therefore, we use reward model performance in the main paper to evaluate preference datasets.

Table 17: Comparison of DPO-trained models using different human-LLM preference mixes.

| | Downstream Task Performance | | | | | | | | | | |
|---|---|---|---|---|---|---|---|---|---|---|---|
| **Pref. Mix** | MULTIPREF (Appendix A.1) % Direct Human for Best Hybrid: **37.4%** | | | | | | Helpsteer2 (Wang et al., 2024b) % Direct Human for Best Hybrid: **69.6%** | | | | |
| | Avg. | GSM8K | BBH | IFEval | Codex | AlpacaEval | Avg. | GSM8K | BBH | IFEval | Codex | AlpacaEval |
| Best Hybrid | **56.67** | **68.61** | 65.09 | 49.54 | **79.59** | 20.53 | 56.09 | 65.73 | 65.29 | 58.96 | 75.13 | 15.34 |
| 100% Human | 54.93 | 67.10 | 65.06 | 48.06 | 77.95 | 16.48 | 55.83 | 65.13 | 64.97 | 56.56 | 77.89 | 14.59 |
| 75% Human | 54.25 | 66.19 | 65.11 | 47.87 | 74.90 | 17.20 | **56.44** | 65.73 | 65.32 | 56.56 | **79.06** | 15.52 |
| 50% Human | 55.59 | 67.32 | **65.80** | **50.83** | 77.37 | 16.63 | 55.60 | 65.01 | 57.67 | 74.42 | **15.93** | |
| 25% Human | 56.15 | 67.70 | 65.26 | 50.09 | 78.53 | 19.14 | 56.25 | **65.81** | 64.77 | 58.23 | 76.53 | 15.91 |
| 100% Synth. | 56.37 | 67.70 | 65.09 | 50.65 | 77.74 | **20.68** | 55.79 | 64.90 | **65.34** | **59.33** | 75.39 | 14.01 |
| BASE SFT | 52.53 | 64.14 | 63.51 | 47.13 | 77.53 | 10.32 | 52.53 | 64.14 | 63.51 | 47.13 | 77.53 | 10.32 |
| **Pref. Mix** | AlpacaFarm (Dubois et al., 2023) % Direct Human for Best Hybrid: **67.2%** | | | | | | ChatArena (Zheng et al., 2023b) % Direct Human for Best Hybrid: **23.0%** | | | | |
| | Avg. | GSM8K | BBH | IFEval | Codex | AlpacaEval | Avg. | GSM8K | BBH | IFEval | Codex | AlpacaEval |
| Best Hybrid | 54.07 | 63.68 | **64.58** | 51.20 | **74.46** | **16.40** | **56.75** | **68.76** | 65.49 | **56.19** | 77.06 | 16.24 |
| 100% Human | 53.71 | 65.05 | 63.97 | **54.34** | 72.89 | 12.29 | 55.32 | 66.87 | 65.24 | 54.34 | 77.29 | 12.84 |
| 75% Human | 53.02 | 63.84 | 63.92 | 53.05 | 71.54 | 12.77 | 56.20 | 67.02 | 65.29 | 55.45 | **78.66** | 14.58 |
| 50% Human | **54.09** | 65.50 | 64.43 | 52.13 | 72.82 | 15.57 | 56.17 | 67.55 | **65.57** | 56.01 | 77.07 | 14.66 |
| 25% Human | 53.88 | **65.58** | 64.26 | 51.39 | 74.19 | 13.98 | 55.55 | 66.41 | 65.17 | 53.79 | 77.81 | 14.57 |
| 100% Synth. | 53.17 | **65.58** | 64.43 | 53.97 | 71.02 | 10.86 | 56.11 | 68.46 | 65.17 | 56.01 | 74.37 | **16.53** |
| BASE SFT | 52.53 | 64.14 | 63.51 | 47.13 | 77.53 | 10.32 | 52.53 | 64.14 | 63.51 | 47.13 | 77.53 | 10.32 |

## A.12 REWARD MODEL TRAINING DETAILS

For all the reward model training experiments in this work, we finetune from the TÜLU-2 13B SFT model introduced in (Ivison et al., 2023). We use a fixed set of hyperparameters listed in Table 18 to conduct the training.

| Hyperparameter | Value |
|---|---|
| Data Type | bf16 |
| Number of Epochs | 1 |
| Optimizer Type | AdamW |
| Weight Decay | 0.0 |
| Learning Rate | 1e-5 |
| End Learning Rate | 1e-6 |
| Warmup Ratio | 0.03 |
| Accumulate Gradient Steps | 4 |
| Sequence Length | 4096 |
| Batch Size | 128 |

Table 18: Reward Model Training Hyperparameters

