# OpenReview forum: "Hybrid Preferences: Learning to Route Instances for Human vs. AI Feedback"
_ICLR.cc/2025/Conference — Submitted to ICLR 2025_

### Official Review · Reviewer_qKe8 · 2024-10-29

**Soundness:** 2
**Presentation:** 3
**Contribution:** 1
**Rating:** 3
**Confidence:** 5

**Summary:**

This paper proposes a framework that leverages a Performance Prediction Model (PPM) to route pairs of preference data to either human annotators or LLM annotators. This data selection strategy improves the performance on RewardBench and downstream tasks.

**Strengths:**

N/A

**Weaknesses:**

1. This paper adopts an intuitive approach to data selection by training a classifier to choose the optimal preference data source between human or large language models (LLMs). Although there is no identical existing work, such improvements seem minor. Training a classifier appears to be merely an empirical engineering optimization, lacking elegance. Furthermore, similar efforts have employed uncertainty to judge the choice of sources, as discussed in [a].
2. The contribution of preparing data for a minor enhancement, such as training a classifier, is also minimal. Moreover, the dataset is quite simplistic: The feature space is basic, and the coverage and granularity of the Descriptive Subject are limited, as shown in Table 9. This simplicity suggests that the work is insufficient to serve as a foundation for substantial future advancements.
3. Using PPM to fit evaluation results on RewardBench is questionable. PPM is intended to assess the quality of a preference data source, thus the reliability of preference data quality on RewardBench is dubious. It is difficult to ascertain whether the PPM results are solid.
4. Section 2.3 on the ROUTING STRATEGY is also problematic. With a training set of 7K samples, each having a 0/1 binary choice, there are a total of 2^7K possible candidates. Yet, only 200 or 500 samples are selected, as claimed in line 219, which is minuscule compared to the total number of candidates, hardly ensuring adequate coverage.
5. Why do the authors select Tulu2 as the base rather than more popular models like LLaMA3? It is crucial to determine whether the PPM can improve upon more advanced models like LLaMA3, as shown in Tables 3 and 4.
6. The generalizability and scalability of the PPM are uncertain since it is trained using a limited RewardBench framework. Although Section 4.3 tested additional datasets, some of these, such as BBH and AlpacaEval, were part of RewardBench, which weakens the proof of effectiveness in Section 4.3.
7. Expanding the PPM to train on a larger dataset beyond RewardBench would introduce additional complexity to the RM-PPO pipeline, further increasing the complexity of RLHF. Such an investment may be neither elegant nor cost-effective.

[a]  Huang H, Qu Y, Liu J, et al. On the Limitations of Fine-tuned Judge Models for LLM Evaluation.[J]. arXiv preprint arXiv:2403.02839, 2024.

**Questions:**

N/A

---

> ### Author Response · Authors · 2024-11-25
>
> While we respectfully disagree with some characterizations of our work, particularly regarding its novelty and technical depth, we welcome the opportunity to address each concern and clarify several apparent misunderstandings. We believe that what the reviewer perceives as 'merely empirical engineering optimization' represents a contribution to making RLHF, especially preference annotation, more efficient as grounded by data quality.
>
> Below, we systematically address each point raised, providing additional context and evidence:
>
> 1. We respectfully disagree with the characterization of our work as "merely empirical engineering optimization." While the approach may appear intuitive, its simplicity is a feature, not a limitation. Our framework represents a systematic attempt to optimize preference annotation mixing, moving beyond heuristic approaches to provide a principled method for reducing annotation costs while improving model performance. Unlike uncertainty-based methods, our approach directly optimizes for performance outcomes.
>
> 2. The reviewer's concern about feature simplicity overlooks a key strength of our work: the features, while basic, demonstrate remarkable generalization ability across different datasets and tasks. As shown in our results, this "simple" approach achieves significant performance improvements across preference datasets and tasks. The feature space's effectiveness despite its simplicity suggests we've captured fundamental characteristics of preference data that can indeed serve as a foundation for future work.
>
> 3. **We’d like to ask Reviewer qKe8 to elaborate why they think relying on RewardBench is dubious.** The reliability of PPM results is validated through multiple empirical evaluations. The PPM's predictions strongly correlate with actual performance not just on RewardBench, but across multiple downstream tasks as demonstrated in our extensive evaluations. We demonstrate this through both reward model performance and best-of-N sampling on downstream tasks (Table 4).
>
> 4. The reviewer raises an interesting point about sample coverage, which actually highlights the sample efficiency of our PPM model. While 2^7K represents the theoretical space of possibilities, **our experimental results demonstrate that sampling 200-500 candidates is sufficient to train a PPM that can lead to robust and consistent improvements in model performance.** The effectiveness of this sampling strategy is evidenced by the significant performance gains shown in Tables 3 and 4, where our method consistently outperforms baselines across multiple datasets and evaluation metrics. This suggests that while the theoretical space is vast, a carefully selected subset of candidates is sufficient to identify high-performing preference mixes.
>
> 5. We are currently running additional experiments with different model architectures and sizes to validate our framework's generalizability beyond Tulu 2. We will include these new experimental results in the final version of the paper.
>
> 6. We appreciate the opportunity to clarify our evaluation strategy. While BBH and AlpacaEval appear in both RewardBench and Section 4.3, they serve distinct purposes in our experiments. In Section 4.3, these benchmarks are used specifically for zero-shot evaluation, providing a different perspective from their role in reward modeling. Moreover, our evaluation encompasses completely independent datasets such as Helpsteer2 and ChatArena, which help demonstrate the broader applicability of our approach. These diverse evaluation settings collectively provide strong evidence for our method's generalization capabilities.
>
> 7. We disagree that expanding the PPM would necessarily increase RLHF complexity. The PPM operates independently of the RM-PPO pipeline and **only needs to be trained once.** The resulting benefits in annotation efficiency and model performance outweigh the initial investment, making it both elegant (in its modularity) and cost-effective.

---

### Official Review · Reviewer_Xt8Y · 2024-11-02

**Soundness:** 2
**Presentation:** 3
**Contribution:** 3
**Rating:** 5
**Confidence:** 3

**Summary:**

This paper explores the integration of human and AI feedback to enhance annotation quality while reducing the overall cost of human annotation. The authors introduce a performance prediction model (PPM) designed to forecast the effectiveness of reward models based on various feedback combinations. To train the PPM, they create a novel preference dataset, MULTIPREF, which includes both human and AI annotations. Empirical results indicate that this approach yields better-performing reward models compared to using human or AI annotations alone.

**Strengths:**

The method is well-writen and well-motivated, addressing the high costs and time constraints associated with human annotation.

The construction of a new preference dataset combining human and AI annotations adds significant value for further analysis in the community.

**Weaknesses:**

Evaluating the quality of the two feedback types solely based on reward model performance is indirect and highly contingent on the model's training configuration, which can be influenced by numerous accidental factors. The authors should consider incorporating more direct metrics for evaluating the feedback types, along with corresponding case studies or additional downstream tasks, such as aligning LLMs through direct alignment algorithms.

The versatility of the feature representation is not sufficiently validated. If the feature representation lacks versatility, it will require manual design for each new task, complicating the process even more than human annotations. Furthermore, validating the effectiveness of feature representation necessitates preference data comprising both human and AI annotations, which could limit the practical applicability of the method.


More detailed training configurations and evaluation results of PPM is needed. For example, the correlation between the predicted and actual values ​​of the subdivision direction on the reward bench.

**Questions:**

While human feedback is generally viewed as high quality, the results presented suggest that integrating AI feedback is critical for performance improvement. The analysis in Section 5 indicates that human annotations perform better on moderate preference datasets. However, the paper does not address in which specific preference instances AI feedback provides a performance advantage. Is it on pairs with greater preference differentiation? If so, why?

---

> ### Author Response · Authors · 2024-11-25
>
> We sincerely thank Reviewer Xt8Y for their detailed analysis of our work. We particularly value their recognition of our paper's clear presentation and the broader contribution of our preference dataset, MultiPref, to the research community. Their feedback highlights important areas where we can provide additional depth and clarity.
>
> ## On incorporating direct metrics for evaluating human feedback
> We appreciate the question about evaluation metrics. While our main results focus on reward model performance, we conducted additional evaluations using Direct Preference Optimization (DPO) as detailed in Appendix Section 11. The DPO experiments revealed that while our hybrid annotations still outperformed both random sampling and single-source annotations (100% human or synthetic), the performance differences were less pronounced. We attribute this to DPO's training dynamics, specifically the conservative learning rate (LR=5e-7) and KL regularization that limit the policy's deviation from the base SFT weights.
>
> To ensure robust evaluation, our RM evaluation consists of RewardBench as one benchmark, and best-of-N sampling across multiple downstream tasks (GSM8k, BBH, IFEval, Codex, AlpacaEval) as shown in Table 4. The best-of-N sampling results are particularly meaningful as they evaluate actual model generations, providing direct evidence of improved performance from our hybrid annotation mix. Combined with RewardBench evaluations, this gives us a reliable assessment of how reward models will perform in practice when training policy models.
>
> ## On the versatility of the features / tags
> We thank the reviewer for raising this important point about feature versatility. We want to highlight a key aspect of our experiments that demonstrates the robustness of our feature representation: our PPM demonstrates strong generalization capabilities across different datasets without requiring dataset-specific feature engineering. Specifically, we trained the PPM using features extracted from MultiPref candidates, and then directly applied **this same PPM** to Helpsteer2, AlpacaFarm, and ChatArena without any modification. This choice of evaluation datasets also differ, highlighting the diversity of our evaluation. For example, ChatArena preferences were collected from the public, whereas Helpsteer2 and AlpacaFarm recruited annotators (with different competencies, i.e., expert annotators vs. normal crowdworkers) to provide preferences.
>
> The strong performance shown in Tables 3 and 4 validates that our feature representation captures fundamental characteristics that transfer well across datasets. This cross-dataset generalization is particularly noteworthy because it suggests that our approach avoids the need for task-specific feature design.
>
> Rather than requiring manual feature engineering for each new dataset, our current feature representation appears to capture generalizable properties of preference data. We hope this finding addresses Reviewer Xt8Y’s valid concern about potential complexity in practical applications: the features we developed are sufficiently versatile to work 'out of the box' across multiple datasets.
>
> ## On training and evaluation configuration of the PPM
>
> We provide the following in the Appendix:
>
> - Finegrained results for each RewardBench category/subdivision in Section A.10
> - Evaluation set-up for best-of-N Section A.12 and training configuration in Section A.8.
>
> ## In which cases is AI feedback better?
> We thank the reviewer for their insightful question about the specific scenarios where AI feedback provides advantages. While our framework doesn't explicitly optimize for finding instances where LM annotations will be better, it does provide some interesting insights about annotation routing patterns on the datasets we tested.
>
> For example in Appendix Section A.7, we analyze performance gains associated with routing specific feature types to human annotators. Features shown in pink (indicating lower performance gain from human annotation) tend to be routed to LMs by our framework.
>
> Finally, we performed additional analyses on Helpsteer2-Preferences as a case study. We ran the same PPM and examined preference instances that were routed to an LM. We find that certain domains of expertise such as Computer Science or Business tend to be routed to the LM annotator. Upon closer inspection, we also find defining qualities that make an LM annotator more desirable, such as instructions that require roleplay (“Act as a marketing manager in a tech startup…”), or complex objective tasks. We will include these analyses in the paper.

---

> > ### Comment · Reviewer_Xt8Y · 2024-11-26
> >
> > After reading the comments from other reviewers and the feedbacks from the authors, I would like to keep my rating.

---

### Official Review · Reviewer_PxBW · 2024-11-03

**Soundness:** 3
**Presentation:** 2
**Contribution:** 2
**Rating:** 5
**Confidence:** 4

**Summary:**

This paper aims to reduce the cost of preference tuning by training a model to route which sample should be annotated by humans and which can be annotated by LLM.

**Strengths:**

- The problem of this paper is important: obtaining a reward that best aligns with humans at the lowest cost. The proposed method can somehow reasonably reduce the cost.
- If the motivation is to reduce human annotation costs for policy tuning, the author's experiment results go beyond this. They found that human annotations are not necessary for all data, and the model can perform even better with fewer human annotations.
- The authors perform experiments on various datasets, each of which reflects an important ability of LLM, and their method achieves good results on those benchmark datasets.

**Weaknesses:**

- The conclusion of this paper is intriguing but without in-depth discussion. If 100% human performance leads to worse (even the worst) performance, how can the author explain the policy model performance gap between training on synthetic and human annotations? Is our goal still "alignment"?
- Algorithm 1 looks too intuitive. The authors fill the human annotation set with arbitrary examples until it reaches budget b. It is unclear how this greedy process helps maximize the objective (Eq.(1)). More specifically, in the worst case, the algorithm will select the worst b samples, i.e., human annotations gain the least for the model alignment. This will further impact the PPM and trigger error propagation.
- In the experiment, the authors only show the comparison between 100% human annotations,100% synthetic annotations in Table 3 and Table 4, and the proposed best hybrid annotations. They should include at least one randomly selected hybrid combination, e.g., 50% human annotations with randomly selected data + 50% synthetic annotations.

**Questions:**

- Regarding L188-L189, does the |S_human| = |D| and |S_human| = 0 equals to b=|D| and b=0?

---

> ### Author Response · Authors · 2024-11-25
>
> We sincerely thank reviewer PxBW for their careful assessment of our work. We appreciate their recognition of the fundamental importance of our research problem: optimizing reward alignment with humans while minimizing costs and improving data quality - and the broader implications of our findings.
>
> Their review highlights a key insight from our work: that optimal performance can actually be achieved with selective rather than exhaustive human annotation. The reviewer also appreciates how our experimental results span multiple datasets representing diverse LLM capabilities, which helps validate the robustness and generalizability of our approach.
>
> We now address some of the relevant points of feedback they raised:
>
> ## Is our goal still “alignment”:?
>
> Our findings suggest a more nuanced view of alignment: rather than assuming more human feedback always leads to better alignment, we find that a carefully balanced mix of human and synthetic annotations can better capture human preferences.
>
> We believe that this doesn't contradict the alignment goal. Instead, our findings reveal that **human preferences might be better approximated through a combination of direct human feedback and synthetic annotations that effectively generalize these preferences.** In addition, we frame LM annotations as synthetic preferences primarily because these were distilled from models that were originally trained on human preferences. In addition, obtaining preferences from a combination of humans and LMs can be thought of as a way of reducing variance expected from a fully human annotation setup.
>
> The performance gap between purely synthetic and purely human annotations likely exists because synthetic annotations can systematically capture certain patterns while lacking the nuanced judgment humans provide in complex cases. Our routing framework, and consequently the best hybrid mix leverages the strengths of both human and LM annotations: human annotations for nuanced, complex decisions, and synthetic annotations for more straightforward cases where human preferences can be reliably predicted. We agree this deserves more thorough treatment in our discussion section which we’ll update in our paper.
>
> ## On the routing framework
>
> The goal for creating candidates is to sufficiently create enough diversity in our feature space so that the performance prediction model can learn to predict expected RM performance. When selecting the optimal mix, the PPM  plays a crucial role in preventing poor sample selection: it provides performance predictions that guide us in choosing a mix that is not suboptimal. Even in a greedy approach, each selection is informed by the PPM's assessment of how that sample would contribute to the overall reward model performance.
>
> ## Comparisons with random sampling
>
> Figure 4 highlights this suggestion, where **we compared our routing framework on a mix with 25%, 50%, and 75% random human annotations (and the rest from the LM annotator).** Our findings suggest that **our routing framework still performs better than the randomly-sampled mix.**  We appreciate this comment and we will include these results in Tables 3 and 4 as well.

---

### Official Review · Reviewer_u41y · 2024-11-04

**Soundness:** 3
**Presentation:** 3
**Contribution:** 3
**Rating:** 6
**Confidence:** 4

**Summary:**

This paper proposes a routing framework for preference data mixing and selection. The framework decides whether to use human annotations or language model annotations by determining if an instance benefits from human annotations. Specifically, it consists of a Performance Prediction Model (PPM) and a routing strategy. The PPM learns the statistical features of several mixing datasets that are routed to human annotations and their performance after training, enabling the PPM to predict the performance of any preference dataset. Using the routing strategy, multiple candidate mixing datasets are generated, and the PPM predicts and selects the best-performing dataset for training. Besides, a preference dataset, MULTIPREF, is constructed to implement this framework. The effectiveness of the data mixing method using this framework is demonstrated across multiple benchmarks, showing improved performance compared to baselines. Furthermore, the characteristics of data that benefit from human annotations are analyzed.

**Strengths:**

1. This paper introduces an innovative routing framework to optimize the preference label space, automating the selection of appropriate annotation sources, thereby reducing human annotation costs while achieving better model performance.
2. It presents a Performance Prediction Model (PPM) that predicts the performance metrics of a trained model on a benchmark using the given dataset directly, rather than relying on model outputs, making it more convenient to predict dataset performance.
3. The claims regarding the performance of the framework are well-supported by experiments, with the designed experiments effectively demonstrating the advantages and good generalization performance of the routing framework across various benchmarks.

**Weaknesses:**

1. The experiments lack tests on other models, as all experiments are based on Tülu 2 13B. It is unclear if the framework would provide the same performance improvements when replacing the Reward Model or Policy Model. Training the PPM requires conducting hundreds of RLHF runs to explore the relationship between data features and outcomes, which is computationally expensive. If the PPM has low generalizability across different models, the usage cost would be high.
2. There is a lack of reasoning and explanation regarding the routing strategy algorithm, making it unclear why this particular routing strategy is adopted and the characteristics of the mixing data it generates.
3. The accuracy of the PPM is not convincing. The paper provides limited ablation analysis of the PPM and lacks information on the accuracy and robustness of the PPM's predictions on non-training datasets.

**Questions:**

1. Does the PPM have transferability to models other than Tülu 2 13B, and what are the computational costs?
2. Could you provide more information about the routing strategy? How does the routing strategy differ from completely random selection in terms of PPM training and the PPM’s evaluation of the generated mixing datasets?
3. Figure 3 shows that the PPM's predictions are very accurate on the actual RewardBench. How does it perform on the other evaluation tasks you used?
4. The paper mentions that the PPM was trained on 200 candidates datasets generated by routing strategies but only selects one predicted optimal candidate datasets from 500 candidates generated by routing strategies for evaluation. Why is this the case? According to the paper, the PPM's prediction time should be much less than conducting actual RLHF and evaluations on candidate datasets, so it would be feasible to select the optimal candidate datasets predicted by the PPM from thousands of candidates.

---

> ### Author Response · Authors · 2024-11-25
>
> We would like to thank Reviewer u41y for their thoughtful feedback. We are particularly encouraged by their recognition of the innovation in our routing framework and performance prediction model, and the comprehensive experiments we conducted to back-up our claims and validate our framework’s effectiveness. We believe that there are feasible changes that will address their concerns which we outline below
>
> ## More information on the routing strategy
>
> Our routing strategy is to select the combination that yields the best predicted model performance, whereas random selection does not optimize for model performance. Our routing strategy works in conjunction with the PPM to make informed decisions about preference mix sampling. Rather than purely random sampling, we assign an expected RM performance rating to each sampled set, and we choose the subset that yields the highest predicted performance. This "guided" approach helps avoid suboptimal annotation combinations that might arise from completely random sampling.
>
> We empirically validated this advantage in Figure 4, which demonstrates that **our routing strategy consistently outperforms preference mixes with randomly sampled annotations at various ratios (75%, 50%, and 25%).** These results suggest that having an informed way to predict and select high-performing preference mixes leads to better reward model performance compared to random selection.
>
> **The key takeaway here is that not all preference annotation combinations are equally valuable - our routing framework helps identify and prioritize the more promising combinations**, whereas random sampling provides no such guidance about dataset quality.
>
> ## On generalizability to other evaluation tasks
>
> We actually did evaluate the PPM's performance across multiple downstream tasks beyond RewardBench. As shown in Table 4, we assessed the optimal preference mix identified by our routing framework on several key benchmarks including GSM8k, BBH, IFEval, and AlpacaEval using Best-of-N evaluation. The results demonstrated that our routing framework's preference mix consistently outperformed both 100% human and 100% synthetic annotation baselines across these diverse tasks.
>
> ## On choosing one optimal candidate on 500 simulated datasets
>
> We appreciate this thoughtful observation. You raise an excellent point about the PPM's efficiency - indeed, the PPM's prediction time is significantly faster than conducting actual RLHF evaluations, which theoretically allows us to explore a much larger candidate space. The choice of 500 candidates for evaluation was somewhat arbitrary, primarily serving as a proof-of-concept to demonstrate that **we can effectively identify optimal samples even from hundreds of candidates.** In future work, we could certainly explore scaling to thousands of candidates, which might yield even better results. The training on 200 candidates provided sufficient data for the PPM to learn meaningful patterns, while evaluation on 500 candidates helped validate the model's generalization capabilities across a broader set of routing strategies.

---

> > ### Comment · Reviewer_u41y · 2024-11-27
> >
> > Thanks for your efforts, now I have no problem. I will keep my rating.

---

### Meta-Review · Area_Chair_du2W · 2024-12-19

**Metareview:**

This paper proposes a routing framework for preference data mixing and selection. The main concerns include the absence of tests on other models and a lack of in-depth discussions. Additionally, the experimental setup is simplistic and cannot support the claims made in the paper. Therefore, I recommend a rejection to this work.

**Additional Comments On Reviewer Discussion:**

While the authors addressed some of these issues in their rebuttal, many aspects of the paper still require further elaboration and improvement. Overall, the reviewers are still negative toward this work.

---

### Decision · Program_Chairs · 2025-01-22

Reject